# Probiotic Supplementation during the Perinatal and Infant Period: Effects on gut Dysbiosis and Disease

**DOI:** 10.3390/nu12082243

**Published:** 2020-07-27

**Authors:** Elisabet Navarro-Tapia, Giorgia Sebastiani, Sebastian Sailer, Laura Almeida Toledano, Mariona Serra-Delgado, Óscar García-Algar, Vicente Andreu-Fernández

**Affiliations:** 1Grup de Recerca Infancia i Entorn (GRIE), Institut D’investigacions Biomèdiques August Pi i Sunyer (IDIBAPS), 08036 Barcelona, Spain; elisabetnavarrotapia@gmail.com; 2Valencian International University (VIU), 46002 Valencia, Spain; 3Department of Neonatology, Hospital Clínic-Maternitat, ICGON, BCNatal, 08028 Barcelona, Spain; gsebasti@clinic.cat (G.S.); sebastian.sailer34@gmail.com (S.S.); 4Institut de Recerca Sant Joan de Déu, 08950 Esplugues de Llobregat, Spain; mserrad@sjdhospitalbarcelona.org; 5BCNatal, Fetal Medicine Research Center (Hospital Clínic and Hospital Sant Joan de Déu), University of Barcelona, 08950 Barcelona, Spain; lalmeida@sjdhospitalbarcelona.org

**Keywords:** probiotics, gut microbiota, dysbiosis, antibiotic resistance, autoimmune diseases, atopic diseases, necrotizing enterocolitis, pregnancy, fetal microbiota, preterm microbiota, infant microbiota, probiotic safety

## Abstract

The perinatal period is crucial to the establishment of lifelong gut microbiota. The abundance and composition of microbiota can be altered by several factors such as preterm delivery, formula feeding, infections, antibiotic treatment, and lifestyle during pregnancy. Gut dysbiosis affects the development of innate and adaptive immune responses and resistance to pathogens, promoting atopic diseases, food sensitization, and infections such as necrotizing enterocolitis (NEC). Recent studies have indicated that the gut microbiota imbalance can be restored after a single or multi-strain probiotic supplementation, especially mixtures of *Lactobacillus* and *Bifidobacterium* strains. Following the systematic search methodology, the current review addresses the importance of probiotics as a preventive or therapeutic tool for dysbiosis produced during the perinatal and infant period. We also discuss the safety of the use of probiotics in pregnant women, preterm neonates, or infants for the treatment of atopic diseases and infections.

## 1. Introduction

Recent studies have indicated that microbiota colonization of the human body starts during pregnancy, altering the paradigm of the fetus as a sterile organism [1,2,3]. Microbial species such as *Staphylococcus* and *Bifidobacterium* have been identified in the meconium of neonates [4], the placenta (*Escherichia, Shigella*, *Propionibacterium*, and Enterobacteriaceae) [5], and the amniotic fluid (*Streptococcus spp*. (several species) and *Fusobacterium nucleatum*) [6] of healthy pregnancies. The microbiota is established in early pregnancy and varies depending on maternal nutritional habits, infections, and gestational age. Furthermore, the delivery mode as well as breastfeeding or formula feeding strongly influences the abundance and diversity of infant microbiota, which modulates the immune system response. For that, the standard profile of healthy infant microbiota is the fecal microbiota of full-term infants, vaginally delivered and exclusively breastfed. This profile is used as the control group in the majority of studies, although its bacterial composition is not completely defined yet. However, it is important to mention that maternal diet and habits will significantly influence maternal gut microbiota which can lead to regional differences.

Preterm infants show an altered gut microbiota composition, with an increase in *Enterobacteriaceae* [7], showing a higher presence of pathogens such as *Klebsiella* [8] or *Escherichia coli* [9] and a higher risk of necrotizing enterocolitis (NEC) than control infants [7]. *Bifidobacteria* and *Lactobacilli*, which protect against different pathogens and are identified in healthy infants, are reduced in preterm infants [7]. Moreover, the treatment of pregnant women or preterm infants with antibiotics including intrapartum antimicrobial prophylaxis also affects the gut microbiota colonization in the fetus or the infant, increasing the abundancy of *Enterobacteriaceae* [10]. Dysbiosis triggered by these imbalances in the microbiota generates a severe delay of immune system maturation, which can lead to autoimmune and atopic diseases such as asthma, rhinitis [11], and lifelong food sensitization [12]. Moreover, the hygiene hypothesis, which relates exposure to environmental microbiota during infancy with the tolerance of allergic processes and resistance to pathogens in adult life, is also based on the crucial role of microbiota in the development of innate and adaptive immune responses [13]. However, the lack of long-term studies makes it difficult to clarify the long-term benefits or consequences of probiotic administration during the perinatal period.

If one of the main causes of all these alterations is the composition of the human microbiota, the use of probiotics and paraprobiotics as therapeutic tools is attracting great interest, especially in the perinatal period and during childhood. For example, allergic diseases such as eczema or recurrent asthma are directly related to the balance between T-helper type 1- (Th1) and Th2-associated cytokines. Perinatal dysbiosis promotes low levels of the Th1 cytokine response, generating a delay in the maturation of the immune system in children [14]. Some probiotics such as *Lactobacilli* and *Bifidobacteria* are administered to increase the levels of Th1, restoring the balance with Th2 and therefore the immune response [15]. Postnatally, probiotics are mainly transferred from mother to infant by breastfeeding.

In this review, we highlight the multiple roles of probiotics as a therapeutic strategy in the balance between health and disease, focusing on atopic diseases such as allergies, asthma, rhinitis, food sensitization, and infectious diseases such as NEC. Furthermore, we cover the use of probiotics through regulation of gut dysbiosis in preterm infants and mothers or infants treated with antibiotics. We also evaluate the safety of probiotics and paraprobiotics when they are applied in pregnant women and infants.

## 2. Materials and Methods 

The methodology to perform this qualitative systematic review encompasses the following processes included in the preferred reporting items for systematic review and meta-analysis (PRISMA) statement [16,17]: definition of the research question, literature search, data collection, evaluation, comparison and synthesis, as well as critical analysis and findings presentation, showing the strengths and weakness of the studies analyzed (Figure 1). A meta-analysis was not performed due to the experimental design differences observed in the studies based on the use of probiotics in pregnant women, neonates and infants, which would generate an important bias in the statistical results.

A bibliographic search strategy was conducted to identify all studies reporting on the use of probiotics during pregnancy, the neonatal and the infant period highlighting their impact on neonatal and infant health. Perinatal period starts at the 20th week of gestation and ends 4 weeks after birth including pregnancy and the neonatal period. Furthermore, the infant period extends from birth to one year of age. However, studies in infants with a treatment extended until two years of age were also considered in the analyses of the present review in order to evaluate the impact of the probiotic treatments. The electronic databases consulted were PubMed (MeSH), Scopus, Web of Science, and the Cochrane Central Register of Controlled Trials. The following descriptors were used in multiple combinations (as MeSH terms or not) with Boolean operators (AND/OR) (see the Appendix A): Section 3.1. “(pregnancy AND microbiome) AND (colonization OR development)”; Section 3.2. “(Microbiota OR dysbiosis) AND (Intrapartum antibiotic prophylaxis OR neonatal antibiotic therapy OR neonatal antimicrobial therapy)”; Section 3.3. “(Dysbiosis) AND (microbiota) AND (allergy OR disease OR food allergy OR atopic dermatitis OR asthma) AND (*bifidobacterium*)”; Section 3.4. “Probiotics OR *bifidobacterium*) AND (human milk OR breastfeeding OR breast milk) AND (transfer OR translocation)” Section 3.5. “(Food Hypersensitivity OR Food allergy) AND Probiotics AND (*Lactobacillus* OR *Bifidobacterium*)”; Section 3.6. “Probiotics AND (asthma OR wheezing OR rhinitis) AND prevention”; Section 3.7. “(Probiotics OR paraprobiotics) AND (preterm infant OR preterm neonate)”; Section 3.8. “Probiotics AND safety AND (pregnancy OR newborn/infant)”.

Inclusion criteria were papers written in English and Spanish (with no geographical restrictions) published from 1 January 2004 to 15 April 2020; the presence of the selected terms in the title or as keywords; the use of a PICOS (patient, intervention, comparators, outcome, and study design) approach; original research performed in humans; selected experimental designs including clinical trials, case–control, longitudinal cohort, and cross-sectional studies. Sample size ≥10. In the section related to preterm infants, eligible studies had to include exclusively neonates ≤32 weeks or birth weight ≤1500 g. The quality of controlled studies referring to randomized, nonrandomized, and pre-post treatment was critically appraised following the Cochrane Collaboration’s Risk of Bias Tool [18]. Exclusion criteria were non-pregnant adult population, children older than two years of age, and neonates with significant birth defects. Interventions using only prebiotics or immunotherapy were also excluded (see details in the Appendix A).

The selection of original manuscripts started by screening titles and abstracts for inclusion, creating a reference list of relevant papers for the topics explored in this review. Two investigators (A.-F.V. and N.-T.E.) conducted each stage of the studies selection, deleted duplicate inputs and reviewed studies as excluded or requiring further assessment. All data were extracted by one investigator (A.-F.V.) and cross-checked by a second investigator (N.-T.E.). In case of discrepancies in the selected studies, we opted for reconciliation through team discussion. The information obtained from each study was: first author, experimental design, number of participants, and control groups; intervention period (prenatal and/or postnatal); dose/duration and strains of probiotics administered; and main outcomes/findings. The eligibility criteria followed the PICOS approach. Population: pregnant women, newborns and infants; intervention: any doses, strains or species of probiotics administered prenatally and/or postnatally within the first year of life; comparators: placebo or no probiotics; outcome: the primary outcome was allergies or food sensitization. All authors performed a critical appraisal for the studies selected following the inclusion criteria, also analyzing the methodology and key results.

The characteristics and number of participants, as well as the results, including strengths, weaknesses, conclusions, and biases, were evaluated.

However, some bias was expected due to the heterogeneous results observed in the literature selected; the different populations compared; the distinct health conditions; the reduced number of randomized trials in pregnant women; the use of questionnaires to detect allergies; and the small sample size observed in some studies. Finally, the studies indicated in Figure 1 for each section of this review were identified through database searching and other sources. Furthermore, the following studies were evaluated and selected after meeting the inclusion criteria, the application of the exclusion criteria, and an eligibility assessment: Section 3.1 (35); Section 3.2 (15); Section 3.3 (16); Section 3.4 (20); Section 3.5 (12); Section 3.6 (10); Section 3.7 (57); Section 3.8 (21).

## 3. Results

### 3.1. Prenatal Development of the Microbiome and Early Colonization

Human microbiome colonization can be understood as a progressive process. In puberty and adulthood, the microbiota shows a higher diversity than in newborns [19]. Avershina et al. analyzed the microbiome of stool samples from a cohort of 86 mothers and their children, concluding that the personal diversity of microbiota increased according to the age of the subjects. However, interindividual diversity decreased with age, being more individually diverse among newborns [20].

The prenatal colonization in newborns is under study. Some authors indicate that the process may be initiated by microbiota located in the placenta and amniotic fluid. Collado et al. analyzed microbiota from placenta and amniotic fluid samples obtained from 15 full-term neonates born by caesarean section. Their main findings were the low diversity of the microbiota as well as the predominance of *Proteobacteria*. This microbiome showed common features with the microbiome of the meconium in neonates, suggesting that the colonization process was initiated prenatally [5].

Tapiainen et al. analyzed 212 first-pass meconium samples of near-term and full-term newborns, sequencing regions of the bacterial 16S rRNA gene. The most abundant phyla were *Firmicutes*, *Proteobacteria*, and *Bacteroidetes,* with a relative abundance of 44%, 28%, and 15%, respectively. The diversity of the microbiome seemed to be influenced by the home environment but not by perinatal factors, leading to the conclusion that microbiota were not altered by immediate perinatal factors but by maternal factors before and during pregnancy, suggesting a transfer of the microbiome from the uterus to the fetal gut [21]. Moreover, some authors showed that maternal diabetes [22], diet [23,24], prenatal stress [25], and mother’s weight [26,27] had an impact on the early gut microbiome [22,23,24,25,26,27]. However, De Goffau et al. found no evidence of the microbiome in 537 placental biopsies from complicated and uncomplicated pregnancies analyzed using various methods of DNA detection However, *Streptococcus agalactiae*, a pathogen that causes neonatal sepsis, was found in 5% of placental samples [28].

#### 3.1.1. Gut Colonization

It is widely accepted that the early neonatal gut microbiome comes from maternal strains [29]. Makino et al. identified maternal monophyletic *Bifidobacterium* strains in the intestines of 11 out of 12 infants born vaginally [30]. The correlation between the early colonization and the delivery mode has been analyzed by many authors. In gut samples of newborns delivered vaginally, the *Bifidobacterium* genus was predominant (especially species such as *Bifidobacterium longum* and *Bifidobacterium catenulatum*) [31], followed by *Bacteroides* and enterobacteria [32]. The first colonizing bacteria enter the intestine through the oral cavity. In spite of some authors proposing that the birth canal microbiota is the most decisive modulator on infants’ oral and gut microbiota, recent studies suggest that the microbiota of the oral cavity in the neonate might have a prenatal origin, preceding the birth canal exposure [33]. Although an imbalance in the birth canal can affect the neonate’s oral microbiota. Li et al. studied whether vulvar disinfection with povidone iodine had an effect on the neonatal oral microbiota in 30 infants. Their results showed a lower presence of *Lactobacillus* and more opportunistic pathogens such as *Staphylococcus*, *Klebsiella*, and *Escherichia* in the disinfected group compared to the non-disinfected and C-section groups [34].

The diversity of intestinal microbiota was lower in infants delivered via C-section than in vaginally delivered newborns [24,29,30,31,32,35,36,37,38,39,40,41,42,43]. Several authors have indicated that the low microbial diversity in infants born through C-section correlates to a low *Bacteroides* [32,35,38,40] and *Bifidobacterium* ratio [29,30,31,35,36,42]. Korpela et al., in a prospective analysis of 100 Swedish mother–infant pairs, highlighted that the initial microbiome mismatch between bacteria and host in C-section-born infants is gradually offset by the environmental microbiome and postnatal maternal microbiome acquisition [29].

In relation to breastfeeding, some authors revealed that breastfed children had a high presence of *Bifidobacterium* in their gut and a low abundance of *Clostridiales* [35]. Conversely, formula-fed infants showed fewer *Bifidobacteria* and significantly higher proportions of *Bacteroides*, *Clostridium coccoides* and *Lactobacillus* groups [32]. Korpela et al. studied the presence of human milk oligosaccharides (HMO) in mothers with a functional fucosyltransferase 2 (FUT2) allele, and no differences in microbiota composition was observed in the vaginally born infants. However, in C-section-born infants (caesarean section), the functional FUT2 allele partially made up for the lack of microbiome diversity [37]. Similarly, Hill et al. showed that prolonged breastfeeding significantly increased the microbiota diversity of children born by caesarean section after 24 weeks, but had no effect on vaginal delivered infants [39].

The early microbiota are influenced by infant maturity at birth [39,44,45,46]. Chernikova et al. analyzed 78 stool samples from premature infants and 189 samples from full-term infants and found that the extremely premature infants, delivered before 28 gestational weeks (GW), had lower bacterial intrapersonal diversity compared to babies born at 28–32 or 32–37 GW. They also observed that, in preterm infants, the proportion of *Bifidobacterium*, *Bacteroides*, and *Streptococcus* was decreased compared to full-term infants [47]. Forsgren et al. also found that the prevalence of *Bifidobacterium* differed in the gut microbiota between the 34–37 GW and >37 GW groups, with a delayed bifidobacteria colonization in late preterm infants [48]. 

Other factors have been proposed to explain the interindividual variability of the neonatal microbiome. Fallani et al. demonstrated that northern European countries were associated with higher proportions of *Bifidobacterium* in the feces of newborns, whereas more diverse microbiota with more *Bacteroides* were obtained in southern European countries [32]. Martin et al.’s analysis revealed that, after birth colonization, the microbiota can also be influenced by the presence of siblings and type of feeding. Interestingly, *Bifidobacterium breve* or *Bifidobacterium longum* subspecies *infantis* were early colonizers regardless of these factors. They found gender as an unexpected confounding factor, with girls being more quickly colonized by *Lactobacillus* in early life [41].

#### 3.1.2. Respiratory Colonization

The respiratory microbiome has been less studied than the gut microbiome. Shilts et al. analyzed the nasal microbial community in 33 full-term infants, which was dominated by the *Firmicutes*, *Actinobacteria*, *Proteobacteria*, and *Bacteroides* lineages. They also found that the richness was higher in infants delivered by C-section and fed with formula compared to the vaginally delivered and breastfeeding group [49]. Stokholm et al. found no differences in the microbiome of hypopharyngeal aspirates depending on the delivery method [42]. On the other hand, a different study showed a delay in respiratory microbiota development, with a late colonization of commensal bacteria, such as *Corynebacterium* and *Dolosigranulum,* in infants born by C-section [50]. The airway microbiome at birth seems to be similar in preterm and full-term infants. However, infants which developed chronic lung disease showed reduced bacterial diversity at birth [51].

#### 3.1.3. Skin

Recent studies have shown that 84% of healthy neonates had their skin colonized 24 h after birth [52], and the major bacterial growth was produced by coagulase-negative *Staphylococcus*. Newborns’ skin barriers interact with microbiota and express certain antimicrobial peptides, such as cathelicidin antimicrobial peptide LL37, to inhibit *Staphylococcus epidermidis* growth. Thus, the constant host–microbe interaction contributes to the stability of the skin microbiota [53].

Moreover, Soeorg et al. showed that the skin of breastfed preterm neonates admitted to a neonatal intensive care unit was colonized with distinct *Staphylococcus epidermidis* strains compared to those found in breast milk. However, neonates gradually acquired strains genetically similar to those found in breast milk, similar to full-term neonates [54].

### 3.2. Gut Dysbiosis Induced by Antibiotic and Nonantibiotic Medications

Intrapartum antibiotic (IPA) prophylaxis are routinely used in up to 40% of deliveries in group *B Streptococcu*s-positive women to prevent vertical transmission to the newborn and in elective or emergency C-sections [55]. Further, postnatal antibiotics are frequently prescribed in NICUs (neonatal intensive care units) for the prevention and treatment of neonatal sepsis, which causes high morbidity and mortality rates, especially in preterm infants. Antibiotic abuse (particularly, broad-spectrum antibiotics) is associated with bacterial (gut) dysbiosis and increased resistance rates, promoting harmful consequences such as necrotizing enterocolitis (NEC) [56,57,58] or fungal infections such as candidemia [59] . From the 434 screened studies related to this topic, 15 potentially eligible articles met the pre-established inclusion criteria, including the use of 16S rRNA gene sequencing methodology to analyze biological samples (Table 1). All studies reported on the antibiotic-induced impact on the composition of neonatal gut microbiota, but treatment options differed between studies: seven studies had mothers receiving intrapartum antibiotics (IPA) [12,60,61,62,63,64,65] five studies focused on postnatal antibiotic therapy [10,66,67,68,69]. A further two studies assessed pre and postnatal antibiotic therapy: Zou et al. analyzed the effects of prenatal antibiotic exposure and the intensity of prenatal and postnatal antibiotic exposure on gut microbiota of preterm infants [70]. Tanaka et al. analyzed the influence of antibiotic exposure to newborn infants or their mothers on the developmental intestinal microbiota [71]. Arboleya et al. evaluated the effects of IPA and postnatal antibiotics administered in the first week versus the second week of life [72]. Six studies reported a decrease in microbial diversity induced by antibiotics [10,62,64,65,66,71], while three studies showed no statistically significant differences between the antibiotic exposure and antibiotic-free groups [67,69,70]. Low microbial diversity indices were associated with NEC and a high risk of obesity and inflammatory diseases [73]. Regarding microbial composition, eight studies highlighted increased colonization rates of *Enterobacteriaceae* [10,60,62,65,67,69,70,72], a risk factor for necrotizing enterocolitis (NEC) and sepsis, especially in the preterm population. Both IPA and postnatal antibiotic therapy also showed an impact on protective phylum *Bacteroidetes* (eight studies) [60,64,66,67,69,70,71,72], which provides colonization resistance against antibiotic-resistant bacteria and pathogenic bacteria (*Enterobacteriaceae* and *Clostridium difficile*) [74,75]. Further reduced colonization rates of *Bifidobacteriaceae* were reported in twelve studies [60,61,62,63,65,67,68,69,70,71,72,76] and *Actinobacteria* in six studies [60,62,63,64,67,72]. Nine studies showed an increase in *Proteobacteria* [60,62,63,64,66,67,69,70,72] colonization which is a marker of an unstable microbial community, especially when associated with decreased levels of the phylum *Firmicutes* [77]. Moreover, dybiosis induced by antibiotics was also analyzed, detecting more studies which assessed IPA rather than a postnatal antibiotic therapy. In the IPA group the decrease in *Bifidobacteriaceae* colonization was the most common dysbiotic microbiome alteration [61,62,63,64,70,71,72,76] followed by *Proteobacteria* increase [60,62,63,64,65,70], *Bacteroidetes* decrease [60,64,65,70,72], and *Actinobacteria* decrease [60,62,63,64,72]. Increased *Enterobacteriaceae* colonization was reported in the IPA group of three studies [60,62,72]. Postnatal antibiotic therapy was associated with increased *Enterobacteriaceae* colonization [10,67,69,70,71] followed by *Proteobacteria* increase [66,67,69,70] and *Bifidobacteriaceae* decrease [67,68,70,71]. *Bacteroidetes* [66,67,69] and *Actinobacteria* decrease [67] were less frequently reported.

Nonantibiotic medications also showed an impact on the development of the gut microbiota. Nevertheless, there is still a lack of evidence in the neonatal cohort. Le Bastard et al. conducted a systemic review assessing the impact of proton pump inhibitors (PPIs), metformin, nonsteroidal anti-inflammatory drugs (NSAIDs), opioids, statins, and antipsychotics. PPIs and antipsychotic medication decreased, whereas opioids increased microbial diversity indices. PPIs decreased *Clostridiaceae* and increased *Actinomycetaceae*, *Micrococcacceae*, and *Streptococcaceae*, changes associated with gut dysbiosis and risk for *Clostridium difficile* infection. *Enterococcaceae* or Gammaproteobacteria *(Enterobacter*, *E. coli*, *Klebsiella)* counts were not increased [78]. In conclusion, IPA and postnatal antibiotic therapy affect the composition of the neonatal gut microbiota and may increase the risk for NEC and sepsis by *Enterobacteriaceae* predominance and reduction in protective phyla. Therefore, an antibiotic stewardship may be of utmost importance to reduce unnecessary and harmful antibiotic consequences. Future investigations should focus on: (1) the potential long-term effects of neonatal gut dysbiosis, (2) the effects of nonantibiotic medications in neonatal age, and (3) the definition of biomarkers of induced gut dysbiosis.

### 3.3. Early Aberrant Microbiota and Its Effect on Pediatric Diseases

Allergic diseases occur at any stage of life, although some allergic manifestations, such as allergies to food, are most likely to develop during the first few years of life [79]. After three years of age, the prevalence of IgE, specific to inhalant allergen, becomes predominant [80]. Recent findings demonstrated that variables such as the antibiotic consumption during pregnancy, mode of delivery, feeding and mother’s lifestyle during pregnancy strongly influence the neonatal gut microbiome modulating the development and function of the immune system in the host [23,64,81,82]. This microbiome will interact with receptors of the intestinal immune cells, causing the maturation of the intestinal mucosa and its gut-associated lymphoid tissues (GALT) [83]. GALT is functionally connected to the mesenteric lymph nodes and is able to identify pathogens from non-pathogenic microorganisms or antigens and defense against pathogens. Thus, a proper crosstalk between the immune system and microbiota will establish a Th1/Th2 balance, while an early developmental dysbiosis may underlie allergies or intolerances [84,85]. Moreover, the capacity to trigger the Th1 response is specially limited in neonates and infants, due to maternal IgGs in the placental barrier which partially protect the fetus during the last stages of pregnancy [86], and by secretory IgA (SIgA) in breast milk if breastfed. For that, a deficit of human milk intake, which contains several Igs including IgA, SIgA, IgM, secretory IgM, and IgG can produce low levels of SIgA at the intestinal barrier in infants [87]. A low diversity and abundance of bacterial populations promotes an impaired stimulation of SIgA, which targets an extensive number of gut bacteria modulating their growth [87,88] Therefore, reduced diversity of gut microbiota, low levels of mucosal IgA (total) and an aberrant IgA responsiveness to the gut microbiota during infancy are associated with the allergic diseases development [89]. Breast milk contains not only immunological components to protect infants against infections and allergies, but also human milk oligosaccharides (HMOs). These complex sugars stimulate the growth and/or activity of beneficial bacteria such as *Bifidobacterium* [90]. The genus *Bifidobacterium* represents one of the dominant bacterial groups in the gut microbiota during early life due to its ability to metabolize different forms of HMO [91]. In the context of allergic diseases, several studies based on murine and in vitro models have highlighted the potential role of *Bifidobacterium* in reducing inflammation through the production of anti-inflammatory cytokines and suppressing Th2 immune response and IgE production [92,93,94]. Moreover, *Bifidobacterium* produces short-chain fatty acids (SCFA) that decrease intestinal permeability and maintain the integrity of the intestinal barrier [95] to prevent the triggering of the immune response against antigens in the bloodstream [96]. In patients with atopic dermatitis (AD), the proportion of *Clostridia*, *Clostridium difficile*, *Escherichia coli*, and *Staphylococcus aureus* in the gut microbiome is higher than in healthy controls, and a reduction in short-chain fatty acid (SCFA)-producing bacteria (*Bifidobacterium*, Blautia, Coprococcus, Eubacterium, and *Propionibacterium*) is also observed [97,98]. Interestingly, a new observational study with 94,929 children from both genders showed that gastroenteritis (GE) during infancy could affect the intestinal microbiota in early life and increase rates of asthma, allergic rhinitis, and atopic dermatitis in later life (6 months–5 years). However, the authors did not perform a clinical evaluation of biological samples, thus further studies to find the association between early-infectious-GE, early-noninfectious-GE, and allergic disease are needed [99]. In a recent population-based study of 4.7 million people in British Columbia, the authors identified a 260% decrease in asthma incidence between 2000 and 2014 in young children, which correlated with a large decrease in antibiotic prescriptions. Moreover, antibiotic use in the first year of life was associated with around a doubled risk of asthma diagnosis at five years of age. Additionally, the authors identified a decrease in *Faecalibacterium prausnitzii*, *Roseburia*, and *Ruminococcus bromii* and an increase in *Clostridium perfringens*, associated with asthma and antibiotic exposure [100]. All these studies showed how intestinal dysbiosis could be a possible origin of future diseases in later stages of life.

Recently, and for the first time, a human study using culture-independent techniques to investigate the relationship between the mother’s gut microbiota during pregnancy and allergic disease in the offspring showed that maternal carriage of *Prevotella copri* is associated, in a dose–response manner, with a decreased risk of food allergy during infancy [101]. This finding is relevant since, until now, food allergy (FA) has been related to a reduced bacterial diversity and an increased *Enterobacteriaceae* to *Bacteroidaceae* ratio. Vuillermin et al. (2020) presented a new during-pregnancy predictor of food allergy in offspring, *Prevotella copri*, probably associated with its important role in stimulating fetal immune development via the Toll-like receptor 4-dependent pathway and SCFA production [101].

### 3.4. Transfer of Probiotic Bacteria from Mother to Child

Despite human milk being classically considered sterile, irrefutable evidence has demonstrated that human milk contains a diverse bacterial community. Moreover, retrograde transfer of external bacteria into the mammary gland has also a strong role in milk inoculation during lactation [102]. Human milk from healthy women contains approximately 10^3^–10^5^ CFU/mL (where CFU is colony-forming unit) and constitutes one of the main sources of bacteria to the breastfed infant gut. Although culture-dependent techniques have identified some key genera in breast milk, such as *Staphylococcus*, *Streptococcus*, *Lactobacillus*, and *Bifidobacterium* spp., culture-independent techniques, based on the amplification of the gene coding for bacterial 16S ribosomal RNA (rRNA), have allowed a more comprehensive assessment of the bacterial diversity in human milk. Thus, several studies have described a “core” microbiome of breast milk, consisting of *Streptococcus*, *Staphylococcus*, and *Propionibacterium*, although these genera vary depending on the population studied, the hypervariable region selected, and the milk extraction method used [103,104]. Despite this, potentially beneficial genera such as *Lactobacillus* and *Bifidobacterium*, widely used as probiotics in children for a wide variety of conditions, appear in most published studies, regardless of the use of next-generation sequencing (NGS) or culture-dependent techniques [104,105,106,107,108,109]. A high prevalence of these genera is found in the colostrum and milk following vaginal full-term deliveries [110]. Bacterial translocation from the digestive tract has been proposed as a source of bacteria for the mammary gland during late pregnancy and lactation. This route, called the enteromammary pathway, involves dendritic (DCs) and CD18+ cells, which would be able to take up nonpathogenic bacteria from the GI lumen through the tight junctions and, subsequently, carry them to other locations, including the lactating mammary gland, through the lymphatic system. Thus, this pathway implies a close communication between the gut microbiota and the immune system in all its stages [111,112]. Thus, the intake of probiotics during lactation could be a source of these beneficial bacteria in the infant, aiding the maturation of the intestinal epithelium and the neonatal immune system. Several studies have demonstrated the translocation of probiotic bacteria from the gastrointestinal tract to breast milk via the enteromammary pathway. Jimenez et al. (2008) showed that two probiotic strains isolated from human milk, *L. salivarius* CECT5713 and *L. gasseri* CECT5714, were present in the human milk of six out of 10 women after 30 days of oral intake, although *L. salivarius* CECT5713 appeared in a higher proportion [113]. Two years later, Arroyo et al. (2010) showed that not only *L. salivarius* CECT5713 but also *L. fermentum* CECT5716 appeared in milk samples after 21 days of oral intake [114].

Similar results were obtained with the *L. salivarius* PS2 strain, which was detected in 17 of 29 maternal milk samples from women who took the probiotic from week ~30 of pregnancy until birth [115]. This strain also prevented infectious mastitis in this population, selected for having suffered recurrent episodes in previous pregnancies. *Lactobacillus reuteri*, a probiotic widely used to decrease colic in breastfed babies, has also been isolated in the colostrum of women who have taken it during the last four weeks of pregnancy, showing a significant presence compared to the placebo (12% vs. 2%). However, no difference between groups was observed in the prevalence of probiotics in breastmilk one month after delivery [116].

Although all these studies showed an increased presence of the administered bacteria in the breast milk of mothers, several studies did not demonstrate this enteromammary route in the studied strains. Simpson et al. (2018) found that maternal supplementation of *Lactobacillus rhamnosus* GG, *Lactobacillus acidophilus* La-5, and *Bifidobacterium animalis* ssp. *lactis* Bb-12 did not significantly affect the general breastfeeding-associated microbiota at 10 d or three months postpartum after four months of probiotic intake [117]. Based on these results, the authors concluded that strains isolated from breast milk would have a “natural affinity” and a greater ability to be transferred to the milk microbiota. Interestingly, Mastromarino et al. (2015) showed that women with vaginal delivery obtained higher amounts of lactobacilli and bifidobacteria in colostrum and mature milk compared to probiotic supplemented women who had a caesarean section. Thus, the authors established that the type of birth also influences the structure of the milk microbiota [118].

All these results demonstrate that there are many factors influencing the transfer of probiotics from mother to child through breast milk. Although the existence of an enteromammary route has been demonstrated, the strains isolated from breast milk reach the mammary glands more easily than other strains not present in breast milk. Significant differences in immunoregulatory factors such as cytokines and hormones are seen before childbirth in women with vaginal delivery compared to C-section [119]. Furthermore, a significant efflux of intestinal immune cells to the mammary glands during late pregnancy and lactation has been shown [120]. Thus, the type of delivery, as well as the time and duration of the treatment, will determine the greater or lesser translocation of the probiotic from the intestines to the breast milk. Moreover, the variability of sample collection and the DNA extraction and identification techniques of the strains can generate bias in the results. More studies are needed to determine the effects of specific probiotic strains on the breast milk composition.

### 3.5. Probiotics for the Prevention of Food Sensitization in Infants: Administration to Mothers Versus Infants 

Food allergies (FA) have become a common problem that affects approximately 6% of infants under two and 9% of children aged 3 to 5 [79]. Eggs, milk and peanuts are the most common food allergens, and skin problems, such as eczema, are closely associated with FA [121,122]. The use of probiotics to prevent food reactions has gained popularity in clinical practice, considering that the gastrointestinal microbiota may modulate the mucosal physiology, the barrier function, and systemic immunologic and inflammatory responses. For that, the evaluation of probiotic supplementation in the prenatal and/or postnatal stage during the first months of life, based on randomized, double-blind, and placebo-controlled trials, is necessary to provide the latest evidence about food hypersensitivity in young children. A total of 10 studies published between 2004 and 2020 were included in the analysis: five of them referred to the supplementation of probiotics during the prenatal or pre- and postnatal stages (Table 2) and five only during the postnatal stage (Table 3). In reference to probiotic intake during the postnatal stage, four of the five studies failed to obtain significant results regarding the incidence of food allergy or allergen sensitization in children with a cow’s milk allergy (CMA) or with a high risk of allergies. Intervention periods showed differences between studies, from treatments starting at birth [123] to probiotic administrations at ages up to one year, not showing administrations beyond a year and a half of age [124,125,126]. Strains, dose, mode of administration (mixed with water, food, formula, or infant cereals), and treatment period also differed among studies. Only one study performed on newborns with a high risk of allergy showed a significant decrease in sensitization to cow’s milk (CM), due to using for one year a nonhydrolyzed formula fermented with *Bifidobacterium breve* C50 and *Streptococcus thermophilus* 065. A significant decrease in positive IgE tests against other foods (hen’s eggs, codfish, wheat flour, soy flour, and peanuts) was also observed [127]. Interestingly, this study was the longest of the five postnatal studies. In addition, the formula contained nonhydrolyzed milk proteins. Probiotic strains were heat-inactivated after milk fermentation, that is, no live bacteria were used during intervention (named paraprobiotics). The authors attributed its effect to cell wall components such as peptidoglycans, which are thermoresistant and able to activate Toll-like receptor (TLR) 2. TLR2 activates the production of mediators such as IL-6, inducing IgA differentiation from naive B cells [128]. Regarding the effect of probiotic supplementation in both the prenatal and postnatal stages, probiotics were administered during the last weeks of pregnancy (weeks 32–36) and probiotic intake in newborns with a high risk of allergy was extended to six months in three of the four studies [129,130,131]. The exception was the study by Abrahamsson et al. (2007), which was prolonged up to one year after delivery [132]. Remarkably, all studies showed a lower sensitization to common food allergens in the probiotic group compared to the placebo, although only two showed statistical significance [129,131]. Both mothers and infants took the same probiotic mixture in all cases, and in only one study the mother also took probiotics after delivery [130]. Kim et al. supplemented a probiotic mix in Korean infants at high risk of food allergy, concluding that there were no changes in the frequency of positive food antigen-specific IgE sensitization and food allergies. However, sensitization against any one of the common food allergens (egg whites, cow’s milk, wheat, peanuts, soybeans, and buckwheat) appeared to be lower in the probiotic (38.7%) than in the placebo group (51.7%). However, this study was limited by the high drop-out rate: there were only 31 individuals in the probiotic group and 29 individuals in the placebo group that were compared for sensitization and prevalence of food allergy [130]. Interestingly, Kuitunen et al. (2009) observed a significant decrease in atopic sensitization (positive food skin prick test (SPT) response and/or food-specific IgE >0.7 kU/L) in caesarean-delivered children compared to vaginally delivered children [131] after prenatal and postnatal probiotic treatment. Similar to Morisset et al., the authors inferred that the transient protection offered by probiotics against IgE-associated allergic diseases is based on the stimulation of Toll-like receptors. Additionally, in the subgroup of caesarean-delivered children, the authors noticed a delayed increase in bifidobacteria recovery in placebo-treated children. Only one of the treatments in the prenatal and postnatal period was performed with a single strain. *Lactobacillus reuteri* ATCC 55730 showed the ability to significantly decrease the levels of circulating IgE to egg white at two years of age after a prenatal and postnatal treatment in infants at a high risk of allergy [132]. However, *Lactobacillus reuteri* ATCC 55730 had no effect on other food allergens, such as cow’s milk, cod, wheat, peanuts, and soybeans. Interestingly, the authors observed that the effect of the treatment was more pronounced in infants whose mothers (and not fathers) have allergic disease. This highlights the significance of the supplementation to mothers in late pregnancy. Finally, no significant differences were observed in positive food SPT in infants at high risk of allergy after a prenatal treatment with *Lactobacillus rhamnosus* GG (Gorbach-Goldin) during the last month of pregnancy [133].

### 3.6. Probiotics for Prevention of Asthma/Wheezing and Rhinitis: Administration to Mothers Versus Infants

Allergic diseases such as asthma, atopic dermatitis (AD), and allergic rhinoconjunctivitis (ARC) are among the main health problems in children and are particularly abundant in western countries. The prevalence of allergic diseases varies based on the population studied, from 9.5% in asthma up to 10–20% in allergic dermatitis in American children [134,135]. The risk factors are multiple, including parental history of allergies, early childhood allergen exposure, lack of breastfeeding, or an immune predisposition to Th2 [136]. Although many follow-ups of RCTs (randomized clinical trials) revealed a lower risk for eczema after probiotic treatment [137], recent studies regarding its effect in preventing asthma, rhinitis, or wheezing need to be reviewed. The latest evidence from randomized, double-blind, and placebo-controlled trials evaluated the preventive properties of probiotic supplementation in the prenatal and/or postnatal stages in asthma/wheezing and rhinitis. Since Elazab et al. (2013) published a meta-analysis about the effect of probiotics in atopy and asthma in early life [138], our analysis contains all the prenatal/postnatal or postnatal probiotic prevention studies in asthma, wheezing, and rhinitis indexed in Pubmed in the last 10 years. A total of 12 randomized, double-blind, placebo-controlled trials were included in the analysis: eight of them referred to the supplementation of probiotics during the pre- and postnatal stage (Table 4) and four during the postnatal stage (Table 5). Several studies administered probiotics during the first 6 months of life and no study of the postnatal period showed administrations beyond two years of age. Regarding probiotic treatment during the postnatal stage, all of them failed to obtain significant results in the prevention of asthma, rhinitis, or conjunctivitis. Three of the four studies agreed on the duration of treatment (six months), except for the study of West et al. (2013), which extended the treatment to nine months. The age at intervention was also different, as well as the strains, the doses, and the mode of administration (mixed with water, infant cereals, cow’s milk-based formula, partially hydrolyzed whey formula, or breast milk), although half of the studies agreed to start treatment after birth and in newborns with a high risk of allergies [139,140]. A significant difference in the incidence of eczema was observed in the study of Schmidt (2019) compared to placebo according to a previous meta-analysis that showed a significantly lower risk ratio (RR) for eczema compared to controls, especially those supplemented with a probiotic mixture [137]. However, it is important to note that asthma, rhinitis, and conjunctivitis develop later in childhood [141,142]. In Schmidt’s study, the children had a maximum age of 20 months at the end of the treatment and no follow-up was performed, so that the protective effect of the probiotic could have been masked. In the remaining postnatal studies, the authors observed a lower cumulative incidence of asthma and allergic rhinitis in the probiotic groups compared to the placebo group (asthma: 9.7% vs. 17.4% [139]; rhinitis 12.9% vs. 19% [140] at five years of age), but differences were not significant. There was no significant improvement in lung function after a follow-up of 8–9 years in children treated with *Lactobacillus paracasei* F19 for nine months at four months of age [124]. However, this study lost 60–70% of the original population after follow-up, probably biasing the results. In the same line, the intervention with probiotics during the prenatal and postnatal stages did not prevent rhinitis or asthma/wheezing during childhood. Only Wickens et al. observed a significant reduction in the prevalence of wheezing (64.2% vs. 76.8%), eczema (42.1% vs. 59.4%), and atopic sensitization in (49.5% vs. 62.3%) in the group treated with *Lactobacillus rhamnosus* HN001 compared to the placebo after 11 years of follow-up. No statistical significance for rhinitis prevention was obtained after probiotic treatment (65.6% vs. 73.5%). This study is the longest related to these outcomes for now [143]. Interestingly, this same study showed a significantly lower incidence for eczema at four years (32.7% vs. 49.3%), but not for asthma or wheezing [144]. However, despite the high rate of participation at four years of age (about 90% in all three groups), no significant results were obtained for the *Bifidobacterium lactis* HN019 group. One of the eight pre/postnatal studies showed, using electronic follow-up data, a higher prevalence of asthma in the children of the probiotic group versus placebo at five years (31% vs. 17%). These results contradict some meta-analyses that did not find an increased risk of asthma or wheezing [138]. As the authors mentioned, children or breastfeeding mothers could have taken commercial probiotics during the follow-up period [145]. Neither a significantly lower incidence of allergic rhinoconjunctivitis nor asthma/wheezing was seen at two or six years of age after the supplementation of LGG, *B. lactis* Bb-12, and *L. acidophilus* La-5 in mothers for four months. Although AD incidence was significantly lower in both studies [146,147], insufficient statistical power and a high proportion of missing data produced nonsignificant results in asthma and ARC.

### 3.7. The Use of Probiotics and Paraprobiotics in Preterm Neonates

Preterm infants experience a delay in bacterial colonization, usually composed by *Bifidobacterium* and *Lactobacillus*, causing the settlement of pathogenic bacteria. The colonization of pathogenic microbes, which leads to dysbiosis in preterm infants, is linked to the delayed introduction of human milk, early antibiotic intervention, a high rate of caesarean delivery, and total parenteral nutrition [152]. This last factor is associated with a lower abundance of *Bacteroides* and *Bifidobacterium* and a significant loss of biodiversity [153].

Disturbances in the gut microbiota may impair the barrier and immune system, leading to delayed maturation of the humoral immune systems and subsequently to inflammatory reactions. The imbalance between proinflammatory response and insufficient anti-inflammatory protection increases the risk of late-onset sepsis (LOS) and necrotizing enterocolitis (NEC), especially in very-low-birth-weight (VLBW) babies [154]. NEC is a harmful pathology, with a high rate of morbidity and mortality, and occurs prevalently in neonates born weighing less than 1500 g. The pathogenesis may be multifactorial, involving the immune system as a response to an ischemic or infectious insult, with the intestinal microbes playing an important role in the pathogenesis of NEC [155].

Preterm neonates are characterized by immature immune pathways, so they cannot control the extension of pathogenic bacteria. Moreover, the immaturity of the gastrointestinal function, in particular intestinal motility, circulatory regulation, the intestinal permeability barrier, and mechanisms of humoral immune defense, enhance the susceptibility to severe diseases. Supplementation with probiotics may regulate the intestinal microbiota and settle the gut, with beneficial bacteria preventing the development of NEC. Probiotics may act on intestinal permeability, enhance mucosal IgA responses, and increase anti-inflammatory cytokines [156].

Several quantitative nonrandomized studies have demonstrated that a prophylactic probiotic in preterm babies <1500 g is associated with lower mortality and morbidity, including a lower risk of NEC or LOS [157,158,159,160]. However, other studies with a similar design did not find this association, probably due to the multifactorial causes of these pathologies [161,162,163] (Table 6).

Regarding randomized clinical trials, numerous studies have demonstrated the relationship between probiotic administration and low mortality and low risk of NEC [164,165,166]. Other studies found a beneficial effect only on LOS [167]; others did not find any beneficial effects of probiotics in reducing the risk of NEC [168,169,170] (Table 6). The strengths of such studies were the longitudinal design, the inclusion of a large number of preterm neonates, the strict inclusion criteria, the stratification for weight and gestational age, antibiotic therapy, caesarean section, maternal pathology, and all conditions that might impair the establishment of a beneficial and functionally active neonatal intestinal microbiota.

The limitations of these studies were that the authors did not report the comorbidity of other diseases such as the persistence of arterial ductus (also a multifactorial pathology) or the ratio of small for gestational age neonates who are at high risk of NEC [171].

Furthermore, there is a high heterogeneity among studies, which could be produced by differences in eligibility criteria, the use of a variety of probiotic species, and the different protocols of dosage and timing. Therefore, the included studies had different strategies concerning enteral feeds (e.g., breast milk vs. formula) and the use of antibiotics. Previous studies reported that more effective colonization by supplemented agents could start at birth, in an uncontaminated gut environment, leading to improved short- and long-term benefits [172].

The most recent studies support the use of *Lactobacillus* and *Bifidobacterium* combination probiotics as most beneficial for preventing NEC in very preterm neonates [160].

*Bifidobacterium* species digest components of human breast milk, such as human oligosaccharides (HMOs), enhancing their establishment in the infant gut and maximizing nutrient utilization. *Bifidobacterium* may create resistance to potentially dangerous pathogens and also stimulate the improvement of the mucosal and systemic immune systems, which play an essential role in enhancing the development of the preterm gut, preventing NEC and LOS [173].

It is important to clarify that the different results of some studies are linked to critical differences between probiotic strains, whose characteristics such as the ability to modulate immunity and infections will be different. Accordingly, randomized clinical trials such as the large UK multicenter Probiotics in Preterm Infants Study (PiPS), which used the probiotic *Bifidobacterium breve* BBG-001 strain, did not improve the prognosis of NEC or LOS [174].

An updated meta-analysis including placebo-controlled studies to explore the effect of *Lactobacillus* on the incidence of NEC in preterm infants showed a significant reduction in the incidence of NEC (RR 0.34, 95% CI 0.25–0.46; *p* < 0.00001) and death (RR 0.48, 95% CI 0.36–0.64; *p* < 0.00001). No significant difference in the incidence of sepsis was found between the *Lactobacillus* and placebo groups (RR 0.90, 95% CI 0.72–1.12; *p* = 0.34) [175]. However, this study had several limitations, such as different doses, strains, duration of supplementation, and variations in the gestational age or birth weight of preterm infants. Another recent meta-analysis based on 23 RCTs (*n* = 4783) of probiotics in preterm neonates in low- and medium-income countries indicated that probiotics are significantly effective at reducing the risk of all-cause mortality, LOS, and NEC in preterm VLBW [176]. However, nearly 40% of those studies carried a high risk of bias.

Interestingly, a network meta-analysis (NMA) to identify the best prevention strategy for NEC in preterm infants concluded that a probiotic mixture and *Bifidobacterium* more significantly reduced the incidence of NEC than *Lactobacilli*, *Bacillus*, or *Saccharomyces* [177]. The last published guidelines of The European Society for Paediatric Gastroenterology Hepatology and Nutrition (ESPGHAN) recommend that, if all safety conditions are met, the use of *Lactobacillus rhamnosus* GG ATCC 53103 at a dose ranging from 1 × 10^9^ CFU to 6 × 10^9^ CFU might reduce NEC stage 2 or 3, although with low certainty of evidence. Most recently, the panel conditionally recommended using a combination of *Bifidobacterium infantis* Bb-02, *Bifidobacterium lactis* Bb-12, and *Streptococcus thermophilus* TH-4 at a dose of 3.0 to 3.5 × 10^8^ CFU (of each strain) as it might reduce NEC stage 2 or 3 (low certainty of evidence) [178].

Nevertheless, it has been reported that probiotics have the potential to cause probiotic-related sepsis [179], but in the analyzed studies the authors did not find any adverse effect. Accordingly, some authors recommend screening the safety of probiotic supplements for antibiotic resistance in commercially manufactured probiotic supplements [180]. Defined as nonviable microbial cells (intact or broken) or crude cell extracts, paraprobiotics have been proposed as a potential alternative, although more clinical trials on this topic are necessary [181].

### 3.8. Safety of Probiotics in Pregnancy and Neonatal Period

In recent years, probiotics have been routinely used in pregnant women and newborns. Probiotic preparations can be administered alone or in combination with antibiotics, especially for gastrointestinal and genitourinary health. Interestingly, the Food and Drug Administration (FDA) regulated probiotics such as nutritional components; their regulation varies between regions and is focused on the legitimacy of any counter-claims, rather than the efficacy, safety, and quality [183]. Thus, these regulatory deficits may have serious consequences for vulnerable groups, and a careful safety evaluation is required before their use. The European Commission’s Scientific Committee on Food (SCF) recommended the exclusion of *Enterococcus* strains as probiotics [184]. Due to the increased use of probiotics, several studies have focused on the effectiveness and the safety of probiotics in recent years. Of the initial 75 studies selected in this review, after eligibility assessment we included 21 studies that met the inclusion criteria; all clinical trials published in the last 10 years that evaluated the safety of probiotics in pregnancy and the neonatal period.

Only one of the studies reviewed reported adverse effects due to the use of probiotics. Topcuoglu et al. included 210 neonates born before 32 weeks of gestation for a randomized controlled trial of a probiotic preparation containing *Lactobacillus casei*, *Lactobacillus rhamnosus*, *Lactobacillus plantarum*, *Bifidobacterium lactis*, fructooligosaccharide, galactooligosaccharide, colostrums, and lactoferrin (no strain names are published). A vancomycin-resistant *Enterococcus* (VRE) outbreak occurred while the probiotic trial was being conducted. The only difference found in the VRE newborn group was the use of probiotics (*p* < 0.001). The authors concluded that concomitant probiotic and vancomycin treatment increases the risk of developing VRE, probably by the acquisition of resistance genes of bacteria mediated by probiotics use [185]. Two case reports about *Bifidobacterium*-related sepsis have been described in LBW and VLBW neonates, one of them after a surgery on a rare abdominal wall defect (omphalocele) and the other one in an extremely low-birthweight infant (600g) [179,186]. Neither case was life-threatening.

Among the other reviewed studies, no adverse effects were reported from the use of probiotics. The adverse outcomes evaluated were maternal health in one trial [187]; pregnancy-related or fetal adverse outcomes in three trials [187,188,189]; fetal and/or neonatal anthropometry in 10 trials [187,188,190,191,192,193,194,195,196,197]; infections, including severe infections in eight trials [196,198,199,200,201,202,203,204]; allergic disorders in three trials [203,205,206]; gastrointestinal effects in 10 trials [187,190,191,192,193,196,197,200,203,205]; and noncommunicable diseases in one trial [194]. Only five studies evaluated the safety during pregnancy [187,188,189,190,206]. The obstetric variables analyzed by these studies were miscarriage before 22 weeks, caesarean delivery, gestational diabetes, prematurity, fetal growth, Apgar score, birth weight, birth length, head circumference at birth, gestational hypertensive disorders, postpartum hemorrhage, maternal weight gain, and tolerance of the product. For example, the study of Pellonperä et al. on 439 pregnant women evaluated the effect of probiotics on gestational diabetes, observing no differences or adverse effects among groups [189]. Further probiotics behaved as a safe alternative during pregnancy, showing no significant incidence of obstetric complications in the treated groups. Nine studies used single-strain probiotics (*Lactobacillus rhamnosus* GG, *Lactobacillus reuteri* DSM 17938, *Bifidobacterium lactis* CNCM I-3446, *Bifidobacterium longum* subsp. *infantis* CECT7210 and EVC001) [192,193,195,196,197,200,202,203,205] and nine more used a combination of probiotics [187,188,189,190,191,194,201,204,206]. The species of probiotics used most often were *Lactobacillus rhamnosus, Bifidobacterium infantis, Lactobacillus reuteri*, and *Bifidobacterium lactis*. Furthermore, no significant differences were observed between controls and treated groups when safety and adverse effects were analyzed, in spite of some sporadic minor gastrointestinal effects due to specific strains. However, only three of these studies were designed as safety studies [190,194,203]. Furthermore, the sample size was small in some of them, and the probiotic doses were not always the same among the compared studies. Thus, probiotics seem to be a safe alternative in pregnancy and full-term newborns, although more studies designed exclusively to test the safety profile of the studied strains are necessary.

## 4. Discussion 

The colonization of the human microbiome is a progressive process. In recent years, controversy has arisen over whether the colonization starts intrauterine, with the microbiota present in the placenta and the amniotic fluid, or during the delivery process. The available studies agree that the delivery mode modifies the earliest microbiome, with lower proportions of *Bifidobacterium* and *Lactobacillus* and a delayed establishment of gut microbiota in newborns delivered by C-section [207]. Infant maturity at birth also influences the gut early microbiome, with *Bifidobacterium* being one of the genera most affected in late preterm infants [47,48]. The airway microbiome also depends on the delivery mode more than the infant maturity, although further research to clarify this topic is needed. The studies presented in Section 3.2 about IPA and postnatal antibiotic treatment showed a high prevalence of *Proteobacteria* and *Enterococcus* and low levels of *Bifidobacterium*, as well as a delayed colonization of this genus. This early imbalance in the gut microbiota of newborns may be related to future diseases, such as atopic diseases [208]. The combination of prescribed antibiotics and infants born via C-section leads to a larger expression of antibiotic resistance genes in the gut microbiome [209], so the evaluation of the possible overuse of antibiotics in neonatal centers should be performed in future trials.

It is known that the administration of probiotics in early life can stimulate Th1 cytokines to reverse the Th2 imbalance. However, the heterogeneity of studies on the prevention of atopic diseases generates controversy about their effectiveness in prenatal and/or postnatal administration. The few existing studies, together with their heterogeneity and the fact that the properties of one strain cannot be extrapolated to others, make it difficult to establish recommendations for probiotic use for the prevention of food allergies during infancy. Further studies are necessary to verify the specific benefits of individual strains, as several studies have been performed using a mixture of strains. Moreover, differences in the genetic background, mode of delivery, and intestinal microbiota composition among populations could bias the results, limiting the comparability of studies. Despite this, the results suggest that prenatal and postnatal probiotic treatment produce a protective effect on food sensitization compared to the use of probiotics only after delivery. These studies highlight the importance of maternal immunocompetence in utero and breastfeeding microbiota transfer to infants. Furthermore, the study of Morisset et al. highlights the use of heat-inactivated probiotics, also called paraprobiotics, for immune health in infants [127]. Paraprobiotics are inactivated microbial cells (nonviable), are immunologically active, and have been reported to provide health benefits to hosts [210]. Their application in foods could offer interesting advantages such as a longer shelf life, simple storage and transport, easier standardization, and parents’ greater confidence in their use. The World Allergy Organization (WAO) guidelines determined that probiotics could have a benefit in eczema prevention when used in pregnant women. However, studies in breastfeeding women and infants at high risk of developing allergies [211] did not show a significant increase in the prevention of other allergies such as rhinitis, wheezing, or asthma. These results are consistent with a previous meta-analysis [137,138]. However, we considered it necessary in this review to analyze the latest clinical studies not included in previous studies. Only the study carried out by Wickens et al. showed prevention against wheezing, atopic sensitization, and eczema in one of the two strains studied, proving, once again, that the immune-modulating effects of bacteria are strain-specific [143]. Interestingly, previous studies showed that *L*. *rhamnosus* HN001 mostly benefited infants with a genetic predisposition to poor skin, low intestinal barrier function, and an imbalanced Th1/Th2 response [212].

Some points relative to the preventive effect of probiotics on these diseases must be clarified. These allergies usually do not arise until childhood, so longer follow-up studies are necessary. The use of other probiotics or antibiotics during the follow-up period and the high dropout rate, probably due to the preventive effect of treatment, make it difficult to complete the study successfully. The benefits of electronic health records, showed by Davies et al. in terms of a high retention rate after long follow-ups, could be a promising tool to decrease those biases [145]. Moreover, *Lactobacilli* are transient colonizers. Most of the analyzed studies spanned around six months, so it would be desirable to incorporate a longer duration of treatment in further studies. In addition, the delivery mode must be considered as a stratification factor, due to its influence on breastfeeding, infant gut colonization, and allergic risk [213].

Studies have revealed that probiotics are a safe alternative in pregnancy and in full-term newborns. Our analysis detected adverse effects in only three studies, two of them were case studies of bacteremia in VLBW and one after surgery in an LBW infant [179,186]. For that reason, the use of inactivated probiotics in highly vulnerable populations is a promising option for future studies. Probiotics and paraprobiotics are, therefore, a safe therapy with wide applications throughout perinatal life. However, it is imperative to perform rigorous studies and enact regulations to guarantee the safety of the weakest populations. The lack of strict regulations on probiotic manufacturing invites doctors and consumers to demand products whose efficacy and safety have been clearly demonstrated in clinical studies.

## Figures and Tables

**Figure 1 nutrients-12-02243-f001:**
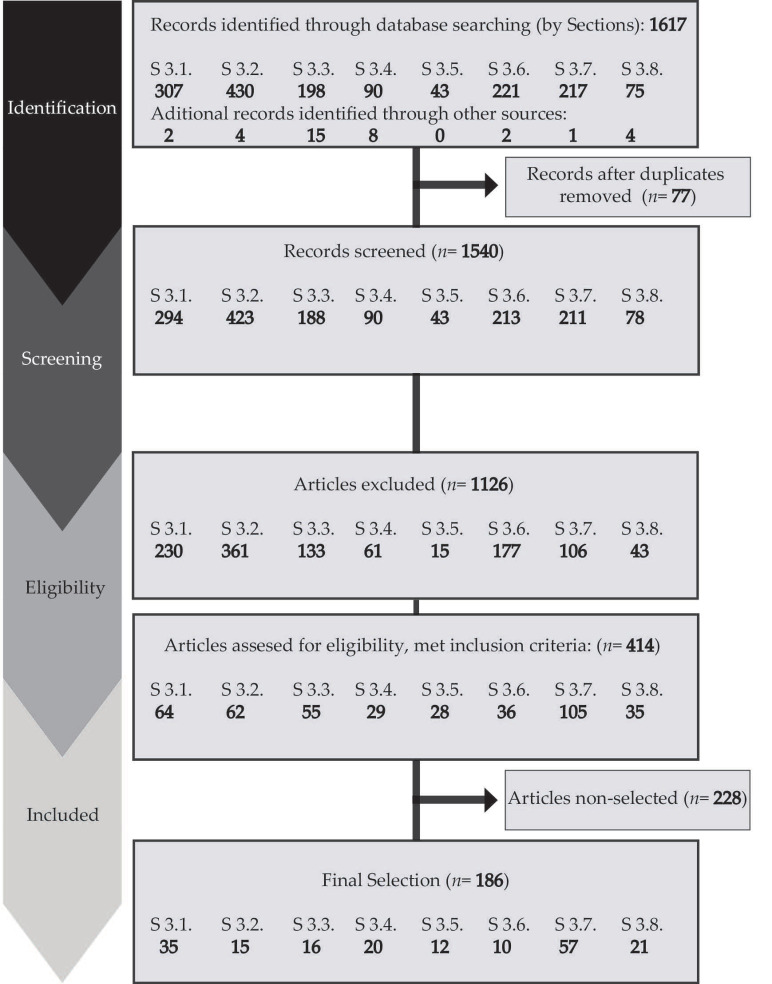
Methodological flowchart following preferred reporting items for systematic review and meta-analysis (PRISMA) for systematic review.

**Table 1 nutrients-12-02243-t001:** Cohort studies of gut dysbiosis induced by antibiotic and nonantibiotic medications.

Author (Year)	Antibiotic Exposure	Objectives	*n*	Population	Key Results	Conclusions
**Studies in Preterm Infants**						
Zou et al. (2018) [70]	Prenatal/ Postnatal	Determine the effects of prenatal antibiotic therapy (PAT) versus prenatal antibiotic free (PAF) group and effects of antibiotic exposure intensity (before and after delivery) on gut microbiota in preterm infants.	24	PAT group (*n* = 12) and PAF group (*n* = 12). Fecal samples on day 7 and day 14. Treatment duration before and after delivery: H-group (>7d) (*n* = 11) versus L-group (<7d) (*n* =11). Fecal samples on day 14.	Phylum level: d7 *Proteobacteria* (PAF 79.75% vs PAT 92.35%) and *Firmicutes* (PAF 9.73% vs PAT 4.69%); d14 *Proteobacteria* (PAF 74.78% vs PAT 87.22%) and *Firmicutes* (PAF 11.19% vs PAT 10.61%). *Bacteroidetes* (PAF 5.75% vs PAT 0.38%). Genus level: d7 *Klebsiella* (PAF 52.17% vs 48.96%), d14 (PAF 45.03% vs PAT 45.81%). *Bifidobacterium* d7 PAT 5% vs PAF 12%. d14 PAT 9% vs PAF 12%. H-group/L-group: Phylum level: *Proteobacteria* (H-group 79.35% vs L-group 70.66%); *Firmicutes* (H-group 19.33% vs L-group 14.81%) Genus level: *Klebsiella* (H-group 55.91% vs L-group 36.15%). *Enterococcus* (H-group 23% vs L-group 34.22%). *Bifidobacterium* (H-group 5.47% vs L-group 10.24%).	The PAT group showed higher prevalence of *Proteobacteria* and significant decrease in *Bacteroides* colonization. Delayed colonization of *Bifidobacterium* in the PAT and H-group. Pre-postnatal antibiotic exposure may affect early gut microbiota composition in preterm infants.
Greenwood et al. (2014) [10]	Postnatal	Determine the impact of empiric ampicillin and gentamicin use in the first week of life on microbial colonization and diversity in preterm infants.	74	Empiric ampicillin and gentamicin. 3 groups: no antibiotics (0d), brief administration (1–4d), intensive administration (5–7d) Fecal samples on w1–w3.	No differences in Simpson diversity index in the first week between groups. Significant decrease in diversity at weeks 2 and 3 in both antibiotic groups (*p* < 0.001 and *p* < 0.004). w1: 0d: *Staphylococcus* 41%, *Enterococcus* 26%, *Enterobacter* 19%. 1–4d: *Enterobacter* 40%; 5–7d: *Enterococcus* 34%, *Clostridium* 33%. w2: *Enterobacter* as the most common genus in patients who received antibiotics in w1. w3: *Enterobacter* (47%), *Enterococcus* (35%) in infants who received intensive administration.	Sustained effects on the gut microbiota by intensive antibiotic therapy in preterm infants. A brief course of antibiotics suppresses the microbiota diversity temporarily.
Arboleya et al. (2015) [72]	Intrapartum/ Postnatal	Assessment of intestinal microbiota in VLBW preterm infants considering perinatal factors as delivery mode and antibiotic use (IPA and postnatal).	40	27 VLBW infants (24–32 WGA) vs. 13 full-term, vaginally delivered, exclusively breast-fed (FTVDBF) neonates without antibiotic exposure. IPA: *n*=14 VLBW vs *n*=3 FTVDBF. Postnatal antibiotics: *n*=12 for 5-8 days after birth, *n*=5 antibiotics starting at 10-13 days of life. Fecal samples: 24hours–48hours, day 10, day 30, day 90	VLBW vs FTVDBF: 24–48h: VLBW group: reduced colonization of *Bacteroidaceae*, *Clostridiaceae, unclassified Actinobacteri* and increased colonization of *Bifidobacteriaceae* and *Lactobacillales* (*p* < 0.05). d10: VLBW group: reduced colonization of *Bacteroidaceae*, *Bifidobacteriaceae* and increased colonization of *Enterobacteriaceae* (*p* < 0.05). d30–d90: increased colonization of *Enterobacteriaceae* and reduced colonization of *Bacteroidaceae* (*p* < 0.05). 30 days of age: infants not exposed to antibiotics showed significantly higher percentages of *Bifidobacteriaceae*, *Streptococcaceae*, and lower of *Enterobacteriaceae* than infants whose mothers received IPA (independently on whether or not the infant received antibiotics).	VLBW group showed reduced *Bacteroidaceae* colonization and increased *Lactobacillaceae* colonization during the first hours of life, followed by a dominance of *Enterobacteriaceae*, on the first days and up to 3 months of age. At 1 month of age, infants whose mothers received IPA had an intestinal microbiota different from that of the infants whose mothers had not received IPA IPA has an equal or higher effect than postnatal antibiotics in the first days of life. Importance of minimizing early medication exposure.
Zwittink et al. (2018) [68]	Postnatal	Effect of postnatal antibiotic treatment duration on preterm gut microbiota.	15	15 late preterm infants (WGA 35.7 ± 0.9) treated with amoxicillin/ceftazidime 3 groups: Antibiotic free (AF) (control): *n* = 5; Short term (ST) (<3days): *n* = 5; Long term (LT) (>5days): *n* = 5. Fecal samples: birth, week1, week2, week3, week4, week6.	AF: high abundance of *Bifidobacterium* w1–w6 (average RA of 73% at w6). ST and LT infants showed significantly lower abundance of *Bifidobacterium* after treatment (*p* = 0.027, 0.027) and at w1 (*p* = 0.027, 0.021), w2 (*p* = 0.016, 0.009) and w3 (*p* = 0.028,0.028) vs. AF. *Enterococcus* dominant in ST and LT infants during w1, not observed in AF group. *Bifidobacterium* abundance significantly decreased until w6 in LT group (*p* = 0.009). *Bifidobacterium* negatively correlated to *Enterococcus*, *Veillonella*, *Clostridium*, *Escherichia*–*Shigella*, and *Enterobacter*.	Short- and long-term treatment with amoxicillin/ceftazidime during the first postnatal week drastically disturbs the normal colonization pattern. ST but not LT allows the recovery of *Bifidobacterium* levels in the first 6 w. *Bifidobacterium* dominance allows higher richness and diversity in gut microbiota.
Dardas et al. (2014) [66]	Postnatal	Determine if the duration of antibiotics within the first 10 or 30 d after birth affects the intestinal microbiome.	29	29 preterm infants (WGA <32) fed with breast milk. G1: 2 days of antibiotic (*n* = 15); G1: 7–10 days of antibiotics (*n* = 12). Fecal samples: 10d and 30d feeds as maternal breast milk and two received exclusively formula.	Significantly lower Shannon–Wiener diversity index in G2 from 10 d samples vs. G1. *Firmicutes* and *Bacteroidetes* dominated the 10d samples, in the 30d samples, the predominant phylum remained *Firmicutes*, but there was a relative rise in *Actinobacteria* and *Proteobacteria*: *Firmicutes* was the predominant phylum.	Rectal microbiota diversity increases over time but decreases with antibiotic exposure. Despite antibiotic pressure, it continues to acquire different bacterial genera.
Zhu et al. (2017) [69]	Postnatal	To assess the effects of one-week antibacterial treatment on the gut bacterial community in preterm infants during the first week of life.	36	36 preterm infants (WGA: 28–37), formula-fed. 3 groups: Penicillin-moxalactam group (PM): *n* = 12; Piperacillin-tazobactam group (PT): *n* = 12; Antibiotic free group (AF): *n* = 12; Fecal samples: day3, day7	No statistical difference in Shannon–Wiener index among groups on both d3 and d7. Significantly lower Shannon–Wiener index in PM (*p* = 0.008) and PT (*p* = 0.028) groups on d7 compared to d3. *Firmicutes* and *Proteobacteria* the most abundant phyla in all groups on d3 and d7. *Bacteroidetes* and *Clostridia* were rarely detected. d3: PT group: *Enterococcus*, *Streptococcus*, and *Pseudomonas* > 60% of the microbiota. *Lactobacillus* significantly higher in PM group (31.57%) than in the other two groups. d7: Higher prevalence of *Bacteroidetes* in PM and PT than in the AF group (*p* < 0.05). Significantly higher prevalence of *Enterococcus* (*p* = 0.003) in PT vs. AF group. Significantly higher prevalence of *Escherichia-Shigella* in the PM vs. AF group (*p* = 0.018).	Prolonged antibiotic therapy affects the early development of gut microbiota in preterm infants. Antibiotic treatment generates a reduction in bacterial diversity and an enrichment of harmful bacteria such as *Streptococcus* and *Pseudomonas*.
**Studies in full-term neonates**						
Nogacka et al. (2017) [63]	Intrapartum	Impact of IPA on the neonatal gut microbiota.	40	IPA group: penicillin (*n* = 18); No-IPA group (*n* = 22). All vaginally delivered full-term babies (>37 WGA). Fecal samples: d2, d10, d30, d90.	*Relative proportion of Proteobacteria:* d2: IPA group: 67% vs. non-IPA 50% 10d: IPA group: 46% vs. non-IPA 35% 90d: IPA group: 34% vs. non-IPA 32% Significantly lower levels of *Bifidobacteriaceae* and *Actinobacteria* (*p* < 0.05) in IPA group.	IPA impacts the establishing neonatal microbiota. The effect remains for at least the first month of life, a very critical time of the development of the microbiota-induced host homeostasis.
Aloisio et al. (2016) [60]	Intrapartum	Evaluate IPA on whole microbiome composition of newborns seven days after birth.	20	10 mothers IPA (ampicillin) versus 10 mothers no IPA. Full-term neonates and vaginal delivery. Fecal samples: d6–d7.	*Actinobacteria*: IPA group 0.4% vs. control 3.8% *Bacteroidetes*: 16% IPA group vs. control 47.7% *Proteobacteria*: IPA group 54.7% vs. control 15.5% (*p* < 0.05) Higher abundance of Gram-negative phyla within the IPA group compared to the control group.	IPA impacts on neonatal gut microbiota reducing microbial biodiversity, allowing colonization of *Enterobacteriaceae*, and reducing the amount of *Actinobacteria*.
Mazzola et al. (2016) [62]	Intrapartum	Assessment of the impact of maternal IPA on the gut microbiota in the first month of life (neonates).	26	4 study groups: 1: Breast-fed infants /control group (BF-C), Group B Streptococcus (GBS) - 2: Breast-fed infants with IPA (BF-IPA), GBS +. 3: Mixed-fed infantes /control group (MF-C) GBS-. 4: Mixed-fed infants with IPA (MF-IPA), GBS+. Fecal samples: d7, d30.	BF-IPA and BF-C: d7: significantly reduced diversity in BF-IPA based on alpha diversity analysis: Chao1 (*p* = 0.0122), Simpson (*p* = 0.035), and Shannon–Wiener (*p* = 0.0082). *Actinobacteria* not detected in BF-IPA, 17% in BF-C. BF-IPA dominated by *Enterobacteriaceae* (*E. coli* 52%). *Bifidobacteria* not detected in BF-IPA. BF-C infants also had higher levels of *Bacteroides*. d30: BF-IPA recovered *Bifidobacteria*; *Enterobacteriaceae* still dominate in BF-IPA infants (44%) vs. BF-C (16%). MF-IPA and MF-C No significant difference in diversity. MF-IPA: increased colonization of *Proteobacteria* (37%) and *Firmicutes* (41%), compared with MF-C. MF-IPA: increased colonization of *Enterobacteriaceae* (35%).	IPA had a significant impact on the early gut microbial composition, which could partially be reversed after 30 days of life.
Azad et al. (2015) [65]	Intrapartum	Assessment of the impact of IPA on neonatal gut microbiota. Secondary objective: assess the role of breastfeeding in modifying antibiotic-induced gut dysbiosis.	198	Full-term neonates, vaginal or C-section birth, and antibiotics. Exposure groups: -no IPA+vaginal delivery; -IPA+vaginal delivery; -IPA+elective CS; -IPA+emergency CS. Fecal samples: m3, m12. Perinatal antibiotics were directly adminis- tered to 8 (4%) infants for suspected sepsis within the first 48 hours after birth, and 69 (37%) of infants received post- natal antibiotics before the 1-year stool collection. Perinatal antibiotics were directly adminis- tered to 8 (4%) infants for suspected sepsis within the first 48 hours after birth, and 69 (37%) of infants received post- natal antibiotics before the 1-year stool collection.	m3: IPA+vaginal delivery was associated with decreased gut microbiota richness (*p* = 0.005). Phylum level: Decreased colonization of *Bacteroides* (24%) compared with 46% among unexposed infants (*p* < 0.05). IPA with CS delivery associated with elevated proportions of *Firmicutes* (*p* < 0.01), and *Proteobacteria* (*p* < 0.05). Genus level: *Enterococcus* and *Clostridium* were predominant. No persistent microbiota differences at one year among infants exposed to IPA with elective CS or vaginal delivery.	IPA in C-section and vaginal delivery are associated with neonatal gut microbiota dysbiosis. IPA was associated with reduced microbiota richness and a depletion of *Bacteroidetes* and increased colonization of *Enterococcus* and *Clostridium*. Breastfeeding modifies some of these effects.
Tanaka et al. (2009) [71]	Prenatal/ Postnatal	Impact of antibiotic treatment in neonates or their mothers on the developmental gut microbiota.	44	*n* = 26: 36–41 WGA Control (antibiotic free—AF) group: *n* = 18 Treatment group (AT): 5 infant subjects were orally administered cefalexin. 3 infants (CD) delivered by C-section: no postnatal antibiotics but their mothers were intravenously injected with cefotiam hydrochloride for 4 days after the delivery. All infants breastfed or given formula. Fecal samples: daily for the first five days and monthly for the first two months.	AT group: diversity decreased from d1 to d3 and remained low until d5. Diversity in AT significantly lower than AF at month 2 (*p* = 0.02). Colonization by *Bifidobacterium* attenuated until one month after birth. High detection rate of *Enterococcus* observed in the AT group since d1 and significantly higher in the first month of life vs. AF (*p* = 0.01). *Enterobacteriaceae* significantly higher in months 1 and 2 (*p* = 0.02) and *Bifidobacterium* count significantly lower on d3 (*p* = 0.03) and d5 (*p* = 0.11) in the AT vs. AF group. AF group: increased colonization during first two months of *Bifidobacterium*, *Clostridium*, *Bacteroidaceae*, and *Veillonella*. *Bifidobacterium* increase from 28% to 67% in the first month. Facultative anaerobes (*Staphylococcus* and *Enterococcus*) did not show such an increasing trend. CD group: reduced intestinal microbiotal diversity compared to AF group. Reduced colonization of *Bifidobacterium* and aberrant growth of *Enterococcus*	Colonization by *Bifidobacterium* was greatly attenuated in both the AT and CD groups. Overgrowth of *Enterococcus* and *Enterobacteriaceae* occurred in most AT infants. Antibiotic administration significantly influences the initial development of the neonatal gut microbiota, with a high impact on *Bifidobacterium* colonization.
Corvaglia et al (2016) [61]	Intrapartum	Effect of IPA on gut microbiota in healthy, full-term infants. Secondary objective: influence of type of feeding on the gut microbiota.	84	84 healthy, full-term infants, born by vaginal delivery. IPA group *n* = 35; No-IPA group *n* = 49 Feeding types: exclusive breastfeeding, exclusive formula feeding, or mixed feeding. Fecal samples: d7, d30	IPA group: significantly lower levels of *Bifidobacterium* at d7 vs. no-IPA group: log CFU/g (5.51–6.98) vs. 7.80 (6.61–8.26) (*p* = 0.000). No significant differences of *Lactobacilli* and *Bacteroides fragilis* at d7 and d30 between groups. No differences in *Bifidobacteria* at d30. Higher counts of *Bifidobacteria* at d7 in no-IPA groups exclusively breastfed. Higher *Lactobacillus* counts both at d7 and d30 in infants exclusively fed human milk, regardless of IPA treatment.	IPA modifies gut microbiota by reducing *Bifidobacteria*, which is further affected in infants receiving formula feeding. Long-term consequences require further investigation.
Aloisio et al. (2014) [76]	Intrapartum	To assess the influence of IPA on the main microbial groups present in the newborn gut microbiota.	52	52 full-term infants, vaginal delivery, exclusively breastfed. IPA group *n* = 26; No-IPA group *n* = 26. Fecal samples: d6–d7.	No-IPA group: *E. coli*, *Bacteroides fragilis* group, and *Bifidobacteria* were the most abundant (9.03, 8.53, and 7.29 log CFU/g, respectively). *Lactobacilli* and *C. difficile* showed lower counts (6.73 and 3.70 log CFU/g, respectively). IPA group: significant reduction in *Bifidobacterium* (from an average of 7.29 to 5.85 log CFU/g). Strong decrement in the frequency of *Bifidobacterium breve*, *B*. *bifidum* and *B*. *dentium* in IPA group. *B*. *pseudocatenulatum*, *B*. *pseudolongum*, and *B*. *longum* less influenced by IPA. *Lactobacillus*, *C. difficile*, and *Bacteroides fragilis* were not significantly affected by IPA.	Significant influence of IPA on the early bifidobacterial pattern of newborns. Further studies are necessary to evaluate the long-term effects of IPA.
Fouhy et al. (2012) [67]	Postnatal	Assessment of consequences after four and eight weeks of postnatal antibiotic treatment within the first 48 h after birth.	18	Treatment group (*n* = 9): combination of ampicillin and gentamicin within 48 h of birth; Control group (*n* = 9): no antibiotics. Fecal samples: w4 and w8 after the end of antibiotic treatment.	week4: Shannon–Wiener index > 3.6 in all samples (high level of biodiversity). Increased *Proteobacteria* colonization (54%) in the treatment group compared to 37% in the control group (*p* = 0.0049). *Bacteroidetes* were detected in less than half of infants treated with antibiotics, notably low levels if present. *Actinobacteria* levels significantly lower in the antibiotic-treated group (3% vs. 24%; *p* = 0.00001). *Enterobacteriaceae* predominant (55% vs. 37%; *p* = 0.0073) and lower levels of *Bifidobacteriaceae* (3% vs. 24%; *p* = 0.0132) in the treatment group. Significantly higher levels of *Bifidobacterium* (25% vs. 5%; *p* = 0.0132) and *Lactobacillus* (4% vs. 1%; *p* = 0.0088) present in the untreated group. week8: significantly higher proportions of *Proteobacteria* (44%) vs. control (23%). *Actinobacteria* increased significantly after four weeks in the treatment group, until there were no significant differences vs. control. *Enterobacteriaceae* decreased after four weeks (*p* = 0.0136) but remained dominant in the antibiotic group (45%).	Postnatal antibiotic therapy induces alterations in the gut microbiota, over eight weeks. The combined use of ampicillin and gentamicin in early life may have significant effects on gut microbiota, but the long-term health implications remain unknown.
Stearns et al. (2017) [64]	Intrapartum	Effects of IPA on the development of gut microbiome among a low-risk population.	74	74 mother–infant pairs IPA group *n* = 21; No IPA group *n* = 53. Fecal samples: d3, d10, w6, w12 postpartum.	Bacterial species richness and Shannon–Wiener diversity index were significantly lower (*p* < 0.01) in infants born vaginally and exposed to IPA at early time points, but reached levels similar to communities in unexposed infants by w12. IPA group: delayed *Actinobacteria* colonization without differences between delivery modes (vaginal/C-section). *Firmicutes* showed delayed colonization in vaginally born infants. Prolonged persistence of *Proteobacteria*. Longer duration of IPA exposure increased the magnitude of the effect on *Bifidobacterium* populations. Infants born by C-section lacked *Bacteroidetes* up to w12 and showed a greater abundance of *Firmicutes*.	IPA affected all aspects of gut microbial ecology including species richness, diversity, community structure, and the abundance of colonizing bacterial genera.

**Abbreviations.** PAT: prenatal antibiotic therapy; PAF: prenatal antibiotic free; H: high time of exposure; L: low time of exposure; RA: relative abundance; WGA: weeks gestational age; IPA: intrapartum antibiotic; VLBW: very low birth weight; FTVDBF: healthy full-term, vaginally delivered, exclusively breast-fed; BF: breast-fed; GBS-: Group B streptococcus negative; GBS+: Group B streptococcus positive; MF: mixed-fed infants; CS: cesarean; AT: antibiotic-treated; AF: antibiotic-free; ST: short treatment; LT: low treatment; PM: penicillin-moxalactam; PT: piperacillin-tazobactam; CFU: colony-forming unit; d: days; w: week; h: hour.

**Table 2 nutrients-12-02243-t002:** Studies focused on the use of probiotics for the prevention of food sensitization in infants.

Source	Intervention Period	Test/Control	Population/Country	Strain(s)/Dose/Administration	Food Allergy-Related Variable	Results
Boyle et al. (2011) [133]	Prenatal only (from the 36th week of pregnancy to delivery)	125/125	Infants at high risk of allergy; Australia	*Lactobacillus rhamnosus* GG (LGG) 1.8 × 10^10^ CFU/day in drops.	Incidence of positive SPTs to food allergens (cow’s milk, eggs, and peanuts) at 12 months.	No significant differences.
Kim et al. (2010) [130]	Prenatal and postnatal (from the 32nd week of pregnancy to six months after delivery)	57/55	Infants at high risk of atopic disease; Korea	Mothers: mixture of *Bifidobacterium bifidum* BGN4; *Bifidobacterium lactis* AD011 and *Lactobacillus* *acidophilus* AD031; 1.6 × 10^9^ CFU/day for each strain for five months. Infants: same mixture from four to six months of age; dissolved in breast milk, infant formula, or sterile water.	Specific IgE against common food allergens (egg white, cow’s milk, wheat, peanuts, soybeans, and buckwheat).	Lower sensitization to any one of the common food allergens in the probiotic group (38.7% vs. 51.7%), but not significant.
Kuitunen et al. (2009) [131]	Prenatal and postnatal (from the 35th week of pregnancy to six months after delivery)	445/446	Infants at high risk of allergy; Finland	Mothers: *Lactobacillus rhamnosus* GG (ATCC 53103); 1 × 10^10^ CFU; *L. rhamnosus* LC705 1 × 10^10^ CFU/day; *Bifidobacterium breve* Bb99 4 × 10^8^ CFU/day, and *Propionibacterium freudenreichii* ssp. *Shermanii* JS 4 × 10^9^ CFU/day) for four weeks. Infants: same mixture for six months; mixed with syrup and 0.8 g of GOS.	Cumulative incidence of any allergic disease and any IgE-mediated allergic disease until age five.	Lower SPT+ and/or food-specific IgE in children born by cesarean vs placebo. No differences in vaginally delivered children.
Allen et al. (2014) [129]	Prenatal and postnatal (from the 36th week of pregnancy to six months after delivery)	220/234	Infants with and without high risk of atopy; UK	Mothers: *Lactobacillus salivarius* CUL61, 6.25 × 10^9^ CFU/day; *L. paracasei* CUL08; *Bifidobacterium animalis* ssp. *lactis* CUL34, and *Bifidobacterium bifidum* CUL20; 1.25 × 10^9^ CFU/day/each strain for four weeks. Infants: same mixture for six months; mixed with breast milk or formula.	Positive SPTs to food allergens (cow’s milk and egg proteins) at either age six months or two years.	Significant decrease in the proportion of SPT+ to CM and eggs in probiotic group after six months; no differences after two years.
Abrahamsson et al. (2007) [132]	Prenatal and postnatal (from the 36th week of pregnancy to 12 months after delivery)	117/115	Infants at high risk of allergy; Sweden	Mothers: *Lactobacillus reuteri* ATCC 55730, 1 × 10^8^ CFU/day in drops; for four weeks. Infants: same mixture for 12 months; mixed with breast milk or hypoallergenic formula.	Incidence of positive SPTs to food allergens (cow’s milk and egg proteins) and specific IgE >0.35 kU/L against common food allergens (egg white, cow’s milk, cod, wheat, peanuts, and soybeans) until two years of age.	Lower incidence of SPT+ to egg in the *L reuteri* group and greater for milk but not significant. Significant lower levels of IgE to egg white in the *L reuteri* group at two years of age.

SPT: skin prick test; AT: α1-antitrypsin; ECP: eosinophil cationic protein; CMA: cow’s milk allergy; GOS: galacto-oligosaccharides; CM: cow’s milk; EBF: exclusive breastfeeding; CFU: colony-forming unit; LGG: *Lactobacillus rhamnosus* Gorbach -Goldin.

**Table 3 nutrients-12-02243-t003:** Studies focused on the use of probiotics in postnatal period for the prevention of food sensitization.

Source	Intervention Period	Test/ Control	Population/ Country	Strain(s)/Dose/Administration	Food Allergy-Related Variable	Results
West et al. (2013) [124]	Postnatal (4 to 13 months of age)	84/87	Healthy, full-term infants with no prior allergic manifestations; Sweden	*Lactobacillus paracasei* F19 1 × 10^8^ CFU/day for nine months; mixed with infant cereals	Specific IgE to cow’s milk, egg white, wheat, codfish, and peanuts after a follow-up of 8–9 years	No significant differences in food allergies compared to placebo.
Taylor et al. (2007) [123]	Postnatal (from birth to 6 months of age)	115/111	Newborns of women with allergy; Australia	*Lactobacillus acidophilus* (LAVRI-A1) 3 × 10^9^ CFU/day for six months; dissolved in 1–2 mL sterile water	Incidence of food allergy and evidence of allergen sensitization (SPT+) after a follow-up of 12 months	No significant differences in the rate of symptomatic food allergy. No significant differences in SPT+.
Viljanen et al. (2005) [125]	Postnatal	88/76/74	Infants with CMA (aged 1.4–11.9 months); Finland	*Lactobacillus rhamnosus* GG (ATCC 53103); 1 × 10^10^ CFU/day or a probiotic mixture (LGG; 1 × 10^10^ CFU/day; *L. rhamnosus* LC705 1 × 10^10^ CFU/day; *Bifidobacterium breve* Bb99 4 × 10^8^ CFU/day, and *Propionibacterium freudenreichii* ssp. *Shermanii* JS 4 × 10^9^ CFU/day) for four weeks; mixed with food	Fecal inflammatory markers as IgA, TNF- α, AT, and ECP	No significant differences.
Hol et al. (2008)	Postnatal	60/59	Infants younger than six months with a diagnosis of CMA; Netherlands	*Lactobacillus casei* CRL431 and *Bifidobacterium lactis* Bb-12; 10^9^ CFU/day for each strain for 12 months; extensively hydrolyzed formula	Clinical tolerance to CM at 6 and 12 months after initial CMA diagnosis.	No significant differences.
Morisset et al. (2011) [127]	Postnatal (from birth until one year old)	66/63	Infants at high risk of allergy; France	Heat-killed *Bifidobacterium breve* C50 and *Streptococcus thermophilus* 065 (4.2 × 10^9^ and 3.84 × 10^7^ bacteria per 100 g of powder formula, respectively). EBF—administered to mothers; No EBF—administered to children (nonhypoallergenic formula)	Incidence of sensitization and allergy to CM and other foods (hen’s eggs, codfish, wheat flour, soy flour, and roasted peanuts) during the first 24 months of life.	Significant decrease in the proportion of SPT+ to CM in probiotic group after 12 months. Significant decrease in positive IgE tests against other foods than CM after 12 months. No significant decrease in incidence of CMA was observed.

SPT: skin prick test; AT: α1-antitrypsin; ECP: eosinophil cationic protein; CMA: cow’s milk allergy; GOS: galacto-oligosaccharides; CM: cow’s milk; EBF: exclusive breastfeeding; CFU: colony-forming unit; LGG: *Lactobacillus rhamnosus* Gorbach -Goldi.

**Table 4 nutrients-12-02243-t004:** Studies based on the use of probiotics during prenatal and pre-postnatal period for the prevention of asthma, wheezing, and rhinitis.

Source	Intervention Period	Test/Control	Population/Country	Strain(s)/Dose/Administration	Allergic Outcome	Conclusions	Risk of Bias
Wickens et al. (2018) [143]	Prenatal and postnatal (from the 35th week of pregnancy to two years of age)	157/158/159	Infants at high risk of allergy; New Zealand	Mothers: *Lactobacillus rhamnosus* HN001; 6 × 10^9^ CFU/day or *Bifidobacterium lactis* HN019; 6 × 10^9^ CFU/each/day for seven months Infants: same mixture for two years	Lifetime prevalence of atopic sensitization, eczema, asthma, wheezing, hay fever, and rhinitis, and relative risks for point or 12-month prevalence at 11 years.	Significant reductions in the 12-month prevalence of eczema and hay fever at age 11 after HN001 supplementation. Significantly lower prevalence of atopic sensitization, eczema, and wheezing in HN001 group. No significant results for HN019	Reduction in participation Rate.
Davies, et al. (2018) [145]	Prenatal and postnatal (from the 36th week of pregnancy to six months of age)	220/234	Healthy infants; UK	Mothers: *Lactobacillus salivarius* CUL61; *L*. *paracasei* CUL08; *Bifidobacterium animalis* ssp. *lactis* CUL34, and *Bifidobacterium bifidum* CUL20; 1 × 10^10^ CFU/day for four weeks Infants: same mixture for six months	Reports of eczema and asthma at five years using electronic follow-up data.	Higher prevalence of asthma in children in the probiotic arm at five years.	Potential intake of probiotics in both groups during follow-up.
Simpson et al. (2015) [147]	Prenatal and postnatal (from the 36th week of pregnancy to three months after delivery)	211/204	Healthy infants; Norway	Mothers: 250 mL of low-fat fermented milk containing *Lactobacillus rhamnosus* GG (LGG); 5 × 10^10^ CFU and *Bifidobacterium animalis* ssp. *lactis* Bb-12 (Bb-12) 5 × 10^9^ CFU and *Lactobacillus acidophilus* La-5; 5 × 10^9^ CFU for four months Infants: no probiotic supplementation	Cumulative incidence of AD and ARC, and the 12-month prevalence of asthma after six years of follow-up.	No significant differences in cumulative incidence of ARC and wheezing at six years of age. Significant differences in cumulative incidence of AD (39.1% in control vs. 29.3% in probiotic group). No statistically significant difference in 12-month prevalence of asthma.	High proportion of missing data.
Abrahamsson et al. (2013) [148]	Prenatal and postnatal (from the 36th week of pregnancy to one year of age)	94/90	Infants at high risk of allergy; Sweden	Mothers: *Lactobacillus reuteri* ATCC 57730; 1 × 10^8^ CFU/day for four weeks Infants: same product for one year	Prevalence of asthma, ARC, allergic urticaria, and eczema after seven years of follow-up.	No significant differences between groups	Significantly greater intake of antibiotic during the first year of life in probiotic group.
Wickens et al. (2012) [144]	Prenatal and postnatal (from the 35th week of pregnancy to six months of age)	157/158/159	Infants at high risk of allergy; New Zealand	Mothers: *Lactobacillus rhamnosus* HN001; 6 × 10^9^ CFU/day or *Bifidobacterium lactis* HN019; 6 × 10^9^ CFU/each/day for seven months (if breastfeeding) Infants: same mixture for two years from birth	Cumulative prevalence of eczema and wheezing occurring between 2–3 months and at age four. Cumulative prevalence of atopic sensitization at two and four years old. RR for the effect of each probiotic on eczema, SCORAD (≥10), wheezing, ARC, and atopic sensitization after four years.	Cumulative prevalence of eczema significantly lower in HN001 group by four years. Some protection against developing SCORAD ≥10, wheezing, and atopic sensitization by age four years but not significant. Significantly reduced risks of having eczema and ARC in the last 12 months at age four. No significant effect of HN019 on any outcome.	Use of antibiotics between two and four years of age significantly higher in the *B. lactis* HN019 group compared to the placebo.
Abrahamsson et al. (2011) [149]	Prenatal and postnatal (from the 36th week of pregnancy to one year of age)	81/80	Infants at high risk of allergy; Sweden	Mothers: *Lactobacillus reuteri* ATCC 57730; 1 × 10^8^ CFU/day for four weeks Infants: same product for one year	Circulating levels of Th1-associated CXC-chemokine ligand CXCL9, CXCL10, and CXCL11 and Th2-associated CC-chemokine ligand CCL17, CCL18, and CCL22 in venous blood at birth, six, 12, and 24 months of age.	Presence of *L. reuteri* in infant stool the first week of life related with low CCL17 and CCL22 and high CXCL11 levels at six months, but no differences in chemokine levels compared to the placebo group.	
Kukkonen et al. (2011) [150]	Prenatal and postnatal (from the 36th week of pregnancy to six months of age)	64/67	Infants at high risk of allergy; Finland	Mothers: *Lactobacillus rhamnosus* GG (ATCC 53103); 1 × 10^10^ CFU/day, *L. rhamnosus* LC705 (DSM 7061) 1 × 10^10^ CFU/day, *Bifidobacterium breve* Bb99 (DSM 13692) 4 × 10^8^ CFU/day and *Propionibacterium freudenreichii* ssp. *shermanii* JS(DSM 7076) 4 × 10^9^ CFU/day for seven months Infants: same mixture plus 0.8 g GOS for six months	Airway inflammation measured as levels of exhaled nitric oxide (FeNO) at age five.	No preventive effect on respiratory allergies. Probiotics had no significant effects on FeNO levels compared to placebo	
Dotterud et al. (2010) [146]	Prenatal and postnatal (from the 36th week of pregnancy to three months of age)	211/204	Infants both with and without a family history of atopy; Finland	Mothers: 250 mL of low-fat fermented milk containing *Lactobacillus rhamnosus* GG (LGG) 5 × 10^10^ CFU and *Bifidobacterium animalis* ssp. *lactis* Bb-12 (Bb-12) 5 × 10^10^ CFU and *Lactobacillus acidophilus* La-5; 5 × 10^9^ CFU for four months Infants: no probiotic supplementation	Diagnosed AD, ARC, or asthma, during the first two years of life.	Significant reduction in the cumulative incidence of AD at two years of age. No reduction in the incidence of asthma or ARC	The nonsignificant results in asthma and ARC may be a result of insufficient statistical power.

Abbreviations: CFU: colony-forming unit; LGG: *Lactobacillus rhamnosus* Gorbach -Goldin; ARC: allergic rhinoconjuntivitis; AD: atopic dermatitis; SCORAD: severity scoring atopic dermatitis; FeNO: nitric oxide.

**Table 5 nutrients-12-02243-t005:** Studies based on the use of probiotics during postnatal period for the prevention of asthma, wheezing, and rhinitis.

Source	Intervention Period	Test/ Control	Population/ Country	Strain(s)/Dose/Administration	Allergic Outcome	Conclusions	Risk of Bias
Schmidt et al. (2019) [151]	Postnatal	144/146	Healthy infants aged 8–14 months; Denmark	*Lactobacillus rhamnosus* (LGG) and *Bifidobacterium animalis* ssp. *lactis* (BB-12) 1 × 10^9^ CFU/each/day for six months; sachet	Incidence of allergic diseases, sensitization, and food reactions.	Significantly lower incidence of eczema in the probiotic group (4.2% vs 11.5%).No differences in the incidence of rhinitis, conjunctivitis asthma, sensitization, or food reactions.	Asthma, rhinitis, and conjunctivitis usually develop later in childhood.Detection of *Bifidobacterium animalis* ssp. *lactis* in the placebo group.
Cabana et al. (2017) [139]	Postnatal (from birth to six months of age)	92/92	Infants at high risk of allergy; USA	*Lactobacillus rhamnosus* GG (LGG); 1 × 10^10^ CFU/day and 225 mg of inulin; dissolve in 2 mL of pumped breast milk, partially hydrolyzed wheyinfant formula, or water	Incidence of eczema within two years of birth and incidences of asthma and allergic rhinitis within five years of birth.	No significant differences in cumulative incidence of eczema (probiotic: 30.9%; control: 28.7%) or asthma (probiotic: 9.7%; control: 17.4%).	Much larger sample size needed to detect a difference in the cumulative incidence of asthma.
Loo et al. (2014) [140]	Postnatal (from birth until six months of age)	124/121	Infants at high risk of allergy; Singapore	*Bifidobacterium longum* BL999; 1 × 10^7^ CFU/g and *Lactobacillus rhamnosus* LPR; 2 × 10^7^ CFU/g; cow’s milk-based infant formula; infants received at least 2.8 × 10^8^ CFU/day	Prevalence of asthma, allergic rhinitis, eczema, and food allergy after five years of follow-up.	No significant differences between groups.	Most of the subjects continued to consume probiotics during follow-up.
West et al. (2013) [124]	Postnatal (from four to 13 months of age)	84/87	Healthy infants born vaginally; Sweden	*Lactobacillus paracasei* ssp *paracasei* F19; 1 × 10^8^ CFU/g; mixed with infant cereals	Prevalence of eczema, allergic rhinitis, asthma, food allergy and lung function after a follow-up of 8–9 years.	No statistically significant differences between the groups.	Loss to follow-up of 60–70% of the original study population.

Abbreviations: CFU: colony-forming unit; LGG: *Lactobacillus rhamnosus* Gorbach-Goldin.

**Table 6 nutrients-12-02243-t006:** Main findings of studies related to the use of probiotics in preterm neonates.

Author, (Year)/Country	Objective	Type of Study, Group (n)	Intervention	Probiotic Strain (Dose)	Primary Outcomes
Luoto (2010) Finland [163]	Evaluation of the impact of the prophylactic use in VLBW preterm infants of *Lactobacillus rhamnosus* GG (LGG) on NEC stage II or III in all five university hospital NICUs in Finland during the VON years.	RCS Prophylactic LGG group (418) Probiotic “on demand” group (1024) CC: 1900	The incidence of NEC was analyzed in <30 weeks or <1500 g babies, from the national database and from the VON databases separately in all five level III NICUs and additionally in three groups according to the probiotic practice.	*Lactobacillus rhamnosus* GG (LGG); 6 × 10^9^ CFU/day	The incidence of NEC was 4.6% vs. 3.3% vs. 1.8% in the prophylactic LGG group, the probiotics “on demand” group, and the no-probiotics group, respectively (*p* = 0.0090). LGG had no influence on the clinical course of NEC.
Braga (2011) Brazil [168]	Evaluation of the combined use of *Lactobacillus casei* and *Bifidobacterium breve* in the prevention of NEC stage ≥2 in VLBW preterm infants.	RDBPC PG: 119 CG: 112	28 days of treatment after second day of life in neonates with a birth weight of 750 to 1499 g.	Multi-strain probiotic: Lactobacillus casei; Bifidobacterium breve, 3.5 × 10^7^ to 3.5 × 10^9^ CFU/day	Confirmed cases of NEC occurred only in the control group (4/112). Infants in PG achieved full enteral feeding faster than CG (*p* = 0.02).
Hunter (2012) USA [158]	Evaluation of the use of *Lactobacillus reuteri* DSM 17938 on the rate of NEC in neonates at highest risk of developing NEC (BW ≤ 1000 g).	RCS PG: 79 CG: 232	Groups separated based on the introduction of probiotic as routine prophylaxis. Treatment from first week of life until hospital discharge.	*Lactobacillus reuteri* DSM 17938; ~5.5 × 10^7^ CFU/day	Significantly lower rates of NEC in the neonates who received *L. reuteri* (2/79 neonates (2.5%) vs. 35/232 untreated neonates (15.1%)). Rates of late-onset Gram-negative or fungal infections were not statistically different between treated and untreated groups (22.8 vs. 31%). No adverse events related to use of *L*. *reuteri*.
Li (2013) USA [162]	Evaluation of the efficacy of probiotic therapy in preventing NEC in VLBW infants.	RCS PG: 291 CG: 289	Screening of patients admitted to the NICU over eight years. Probiotic administration was implemented as part of the standard care for NEC prevention.	Multi-strain probiotic: S. thermophilus; B. infantis; B. bifidum VLBW: 1.05 × 10^9^ CFU/day ELBW: 0.5 × 10^9^ CFU/day	The incidence of NEC was similar between the control group (2.8%) and probiotics group (2.4%) (hazard ratio, 1.15; 95% [CI], 0.42, 3.12). Mortality of NEC similar between groups (1 vs. 2, *p* = 1.000). Incidence of NEC scare was decreased, from 2.8% in the control group to 1.4% in the probiotics group, not significant.
Demirel (2013) Turkey [170]	Evaluation of the efficacy of *Saccharomyces boulardii* for reducing the incidence and severity of NEC in VLBW infants.	Prospective, blinded, randomized controlled trial PG: 136 CG: 135	Treatment from the first feed (within 48 h of birth) until neonates were discharged. The primary outcomes were death or NEC (Bell’s stage ≥2), and secondary outcomes were feeding intolerance and clinical or culture-proven sepsis.	Saccharomyces boulardii; 5 × 10^10^ CFU/day	No significant difference in the incidence of death (3.7% vs. 3.6%, 95% CI of the difference = −5.20–5.25; *p* = 1.0) or incidence of stage ≥2 NEC (4.4% vs. 5.1%, 95% CI, −0.65–5.12; *p* = 1.0) between the PG and CG. Feeding intolerance and clinical sepsis were significantly lower in the probiotic group compared with control (22.9% vs. 29.23% and 34.8% vs. 47.8%).
Fernández-Carrocera (2013) Mexico [165]	Evaluation of the effectiveness of a multispecies probiotic in the prevention of NEC in newborns with birthweight <1500 g.	RDBPC PG: 75 CG: 75	Patients randomized into two groups to receive either a daily feeding supplementation with a multispecies probiotic, 1 g/day, or the placebo. Unspecified treatment period.	Multi-strain probiotic: Lactobacillus acidophilus, 1.0 × 10^9^ CFU/g; Lactobacillus rhamnosus, 4.4 × 10^8^ CFU/g; Lactobacillus casei, 1.0 × 10^9^ CFU/g; Lactobacillus plantarum, 1.76 × 10^8^ CFU/g; Bifidobacterium infantis, 2.76 × 10^7^ CFU/g Streptococcus thermophilus, 6.6 × 10^5^ CFU/g	No differences detected in NEC risk reduction (RR: 0.54, 95% CI 0.21 to 1.39), trend in the reduction in NEC frequency in the studied cases: six (8%) vs. 12 (16%) in the CG. Fewer infants in the PG died or developed NEC vs. CG; RR 0.39 (95% CI 0.17 to 0.87). *Lactobacillus* or *Bifidobacteria* not present in blood cultures in cases of sepsis.
Serce (2013) Turkey [182]	To investigate the efficacy of *S*. *boulardii* in preventing NEC or sepsis in very-low-birth-weight infants.	RDBPC PG: 104 PG: 104	VLBW neonates (BW ≤ 1500 g) treated from the first feed until discharge. The median duration of probiotic supplementation and follow-up was 44 days. The study was conducted in preterm infants (≤ 32 GWs, ≤ 1500 g birth weight). They were randomized either to receive feeding supplementation with *S*. *boulardii* 50 mg/kg every 12 h or a placebo, starting with the first feed and continuing until discharge.	Saccharomyces boulardii, 5 × 10^10^ CFU/day	Same incidence of stage ≥2 NEC in both groups (7/104; 6.7%). No differences between PG vs. CG in late-onset, culture-proven sepsis (18.3% vs. 24.3%, *p* = 0.29); 28.8% vs. 23%, *p* = 0.34), deaths (4.8% vs. 3.8%) or time to reach 100 mL/kg/day of oral feeding (day) (11 ± 7 vs. 12 ± 7, *p* = 0.37).
Bonsante (2013) France [157]	To report outcomes in infants receiving the probiotic cohort (PC) compared with the historical cohort.	RCS PG: 347 CG: 783	Treatment with *Lactobacillus rhamnosus* Lcr35 in neonates born at 24 to 31 weeks’ gestation. Supplementation at the beginning of enteral feeding until a gestational age of 36 weeks or discharge.	*Lactobacillus rhamnosus* Lcr35, 4 × 10^8^ CFU/day	Infants in PG presented a reduced rate of NEC (OR 0.20; 95% CI 0.07 to 0.58), mortality (OR 0.46; 95% CI 0.21 to 1.00), and LOS (OR 0.60; 95% CI 0.40 to 0.89) and achieved FEF significantly earlier (11.7 ± 10 vs. 16.5 ± 13.3; *p* = 0.01). IRB was significantly lower in PG (4.6% vs. 7.5%; *p* = 0.07)
Oncel (2014) Turkey [166]	To evaluate the effect of oral Lactobacillus reuteri in the frequency of NEC and/or death after seven days, frequency of proven sepsis, rates of feeding intolerance, and duration of hospital stay.	RDBPC PG: 200 CG: 200	Treatment with Lactobacillus reuteri DSM 17938 in preterm infants (≤32 weeks). Supplementation started with the first feed and lasted until death or discharge.	Lactobacillus reuteri DSM 17938, 10 × 10^8^ CFU/day (5 drops)	No statistically significant difference between PG and CG in terms of frequency of NEC stage ≥2 (4% vs. 5%; *p* = 0.63), overall NEC, or mortality rates (10% vs. 13.5%; *p* = 0.27). Significantly lower frequency of proven sepsis in PG vs. CG (6.5% vs. 12.5%; *p* = 0.041). Significant difference in rates of feeding intolerance (28% vs. 39.5%; *p* = 0.015) and duration of hospital stay (38 (10–131) vs. 46 (10–180) days; *p* = 0.022).
Janvier (2014) Canada [159]	To determine whether routine probiotic administration to very preterm infants would reduce the incidence of NEC without adverse consequences.	Prospective cohort study, with a historical comparison cohort PG: 264 CG: 317	Treatment with a probiotic mixture as routine administration in preterm infants (≤32 weeks). Supplementation started with the first feed and went until death or 34 weeks postmenstrual age. Comparation with those admitted during the previous 17 months (no probiotic intake).	Multispecies probiotic: B. breve; B. bifidum; B. infantis; B. longum; L. rhamnosus GG (2 × 10^9^ CFU/day)	Significant differences in NEC between PG and CG (5% vs. 10%; *p* < 0.05) and in the combined outcome of death or NEC (11% vs. 17%). No significant differences in death rate between groups. Improvements remained significant after adjustment for gestational age, intrauterine growth restriction, and sex, (OR for NEC, 0.51; 95% CI, 0.26–0.98; OR for death or NEC, 0.56; 95% CI, 0.33–0.93). No probiotic effect on healthcare-associated infection.
Hartel (2014) Germany [161]	To evaluate outcome data in an observational cohort of very-low-birth-weight infants of the German Neonatal Network stratified to prophylactic use of *Lactobacillus acidophilus*/*Bifidobacterium infantis* probiotics.	Observational, prospective, multicentric PG: 2310 CG: 518	Treatment with a probiotic mixture as prophylactic in VLBW infants. Variability regarding dosage and time of probiotic administration: 1 × 1 capsule/day or 2 × 1/2 capsule/day) from day 2 or 3 of life for 14 days or until full enteral feeds. Primary outcome data of all eligible infants were determined according to the center-specific strategy.	Multispecies probiotic: L. acidophilus/B. infantis*	PG associated with a reduced risk for NE surgery (OR 0.58, 95% CI 0.37–0.91; *p* = 0.017), any abdominal surgery (OR 0.7, 95% CI 0.51–0.95; *p* = 0.02), and the combined outcome abdominal surgery and/or death (OR 0.43; 95% CI 0.33–0.56; *p* < 0.001). Probiotics had no effect on the risk of blood-culture confirmed sepsis.
Dang (2015) USA [169]	To investigate the role of probiotics supplementation in improving nutritional outcomes.	RCS PG: 108 CG: 113	Treatment with probiotic mixture as routine administration in preterm infants (≤28 weeks and/or ≤1250g). Supplementation started with the first enteral feeding (48 h of life) and until 34 weeks postmenstrual age. Comparison with those admitted when probiotic intake was not instituted.	Multispecies probiotic: *L*. *rhamnosus* GG (LGG), 5 × 10^8^ CFU/day; *B*. *infantis*, 5 × 10^8^ CFU/day	OR of EUGR significantly lower in PG (–70%): (OR: 0.3, 95% CI: 0.138–0.611). Time to reach full feeds significantly reduced and weight gain significantly better in PG. Significant reduction in number of total parental nutrition days, central line days, nil per os days, and number of feeding intolerance episodes in PG. No significant difference in the incidence of NEC.
Dilli (2015) Turkey [164]	To test the efficacy of probiotic and prebiotic, alone or combined (symbiotic), on the prevention of NEC in VLBW infants.	RDBPC PG: 108 CG: 113	VLBW infant randomized in four groups: G1: probiotic G2: Prebiotic (insulin: 900 mg) G3: Symbiotic: probiotic + prebiotic CG: placebo 1 sachet per day with breast milk or formula until discharge or death, for a maximum of eight weeks.	*B. lactis*, 5 × 10^9^ CFU/day	Significantly lower NEC rate in G1 (2.0%) and G2 (4.0%) groups compared with G3 (12.0%) and placebo (18.0%) groups (*p* < 0.001). Significantly faster times to reach full enteral feeding (*p* < 0.001), lower rates of clinical nosocomial sepsis (*p* = 0.004), and lower mortality rates (*p* = 0.003) in G1, G2, and G3 groups vs. CG. Significantly shorter stays in the neonatal intensive care unit (*p* = 0.002) in G1, G2, and G3 groups vs. CG.
Lambaek (2016) Denmark [167]	To evaluate the benefit of implementing prophylactic use of probiotics as standard care for preterm infants.	Prospective cohort study, with a historical comparison cohort PG: 333 CG: 381	Treatment with a probiotic mixture as routine administration in preterm infants (≤30 weeks). Supplementation started on the third day of life and continued until discharge from hospital. Comparison with a prior period without probiotic use.	Multispecies probiotic: *B. lactis* Bb12, 1 × 10^8^ CFU/day; *L*. *rhamnosus* GG, 2 × 10^9^ CFU/day	Incidence of NEC not significant between groups: (OR) 0.75, (*p* = 0.34, 95% CI: 0.43–1.30). Difference in mortality between groups not statistically significant: OR 0.92 (*p* = 0.55, 95% CI: 0.62–1.40). No side effects; no presence of probiotic strains in blood.
Robertson (2019) UK [160]	To compare the rates of NEC, LOS, and mortality for five-year periods before and after the implementation of routine daily multistrain probiotics administration in high-risk neonates.	RCS PG: 513 CG: 469	Treatment with probiotic mixture as routine administration in preterm neonates at high risk of NEC: Supplementation started on postnatal day 1 and continued until 34 weeks postmenstrual age. Comparison with those admitted when probiotic intake was not instituted.	Multispecies probiotic: Mix 1: Lactobacillus acidophilus and Bifidobacterium bifidum, 1 × 10^9^ CFU/day, each strain Mix 2: *L*. *acidophilus*; *B*. *bifidum*; and *B*. *longum* subsp *infantis*, ~0.5 × 10^9^ CFU/day, each strain The mix used depended on the time of the study.	Rates of NEC significantly decreased from 7.5% (35/469) in CG to 3.1% (16/513) in PG (*p* = 0.014). Cases of LOS significantly decreased from 106/469 (22.6%) in CG to 59/513 (11.5%) in PG (*p* < 0.0001). All-cause mortality decreased from 67/469 (14.3%) to 47/513 (9.2%), although not significant. No episode of sepsis due to *Lactobacillus* or *Bifidobacterium*.

Abbreviations: CC: control group; PG: probiotic group; NICU: neonatal intensive care units; VON: Vermont Oxford Network; RDBPC: randomized double blind placebo control trial; BW: birth weight; VLBW: very low birth weight; RCS: retrospective cohort study; ELBW: extremely low birth weight; OR: odds ratio; LOS: late-onset sepsis; FEF: full enteral feeding; IRB: isolated rectal bleeding; EUGR: extra uterine growth restriction; CI: confidence interval; CFU: colony-forming unit; LGG: *Lactobacillus rhamnosus* Gorbach -Goldin. *No probiotic strains or dose indicated.

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
