# Peer review of "Probiotic Supplementation during the Perinatal and Infant Period: Effects on gut Dysbiosis and Disease"

_nutrients, 2020, doi:10.3390/nu12082243_

Round 1
Reviewer 1 Report
In the current manuscript the authors aim to summarize the body of data related to the therapeutic importance of probiotics in prevention of dysbiosis at the perinatal period. The general idea of this research is interesting, especially as there is a lot of studies and review articles that summarize data for the postnatal and further age from the development of the neonatal. The focus of this review on the perinatal development period makes it unique and interesting. But the manuscript lacks a focus related to the general aim, driven also by the manuscript title.
Points that need to be taken in consideration:
- The authors focus their research not only on the perinatal period of neonatal development but also at postnatal and further development. Therefore, is on one side confusing but also hard to get into the flow and make conclusions. Important is from the beginning to specify what represents the perinatal period (according to your understanding), this is very relevant information, currently missing, that might guide you in structuring of the manuscript.
- It is clear that medication impact the microbiome, especially those with oral intake. Two comments in this aspect, first as presented here it is not clear how this relates to dysbiosis in the perinatal period, and second did the author considered into their selection criteria for this point to exclude studies where the drugs were not delivered orally? Also, the respective table (Table 1) is very confusing as it also includes both intervention with preterm infants or studies on the impact of drugs in postnatal period (more than a week).
- Similarly, to the previous most of the information, with small exceptions (Ref 88), presented in the section 3.2. Early Aberrant Microbiota and its effect on pediatric disease, is difficult to link to the general aim.
- Use research papers and less review papers, e.g. line 237 – Ref 53 and 53, line 278-283 is justified with only one reference (Ref 73), which all are actually review papers. Moreover, general statements as for examples those at line 143-144, line 166-167 need to be supported with respective references.
- Not agree that neither Prevotella (line 171) nor Clostridium coccoides (line 182-183) are pathogens.
- Another issue is related to the statement related to the exclusion criteria for the included studies. Mentioned in line 109 that interventions using probiotics with prebiotics were excluded. However, if you follow the studies summarized in Table 2 – Ref 117 where the probiotic mixture was provided to the infants mixed with syrup and GOS, the last is a prebiotic. In the studies summarized in Table 3, Ref 125 – probiotic was mixed with inulin (a well-known prebiotic); Ref 138 – probiotic was provided to the infants in mix with GOS (prebiotic).
- The reviewer suggests the authors to pay a better attention on what microbiota has been investigated in certain reference and how it fits with their report. In the reference discussed in line 169-172 – Ref 28, the neonatal oral microbiome has been investigated. How this fit with the Gut colonization section?
- Regarding the sections 3.3 and 3.4 that are actually an important part of the manuscript, need to be revised to fit with the perinatal period. Number of studies reported in the respective tables include report data for the postnatal period. The current tables are very busy, removing the postnatal data will make the tables more clear.
- Keep all the table in the same style – the style of Table 1 and Table 2 is different than that of Table 3 and Table 4.
- It is necessary to describe all abbreviations used in the text and the tables as well.
- There is need to improve the numbering of the sections. At the moment there are two sections with number 3.1 – Prenatal development of the microbiome and early colonization; and Gut dysbiosis induced by antibiotic and nonantibiotic medications. Also, not clear why there is a 3.1.1. Early colonization.
Author Response
In the current manuscript the authors aim to summarize the body of data related to the therapeutic importance of probiotics in prevention of dysbiosis at the perinatal period. The general idea of this research is interesting, especially as there is a lot of studies and review articles that summarize data for the postnatal and further age from the development of the neonatal. The focus of this review on the perinatal development period makes it unique and interesting. But the manuscript lacks a focus related to the general aim, driven also by the manuscript title.
Thank you for the recommendation. As reviewer indicated, the manuscript focused on perinatal and infant period from pregnancy to infants with 1 year of age in spite of the follow-up of some of some studies extends up to two years of life from birth. Following your recommendation, we modified the title in order to be more precise in the period addressed (perinatal and infant to 1 year of age) as follow:
Probiotic supplementation during the perinatal and infant period: Effects on gut dysbiosis and disease
As mentioned, a few studies extended the follow-up until 2 years of age from birth (i.e. Wickens et al 2018, table 4). However if the studies are included in this review is due to we consider their results and/or conclusions relevant by the analysis of probiotics use during perinatal or infant period. For example, Wickens (2012) published an article with a follow-up of six months after probiotic treatment so that we decided to include the second article (2018) with a follow-up of two years to analyze the results.
Points that need to be taken in consideration:
1. The authors focus their research not only on the perinatal period of neonatal development but also at postnatal and further development. Therefore, is on one side confusing but also hard to get into the flow and make conclusions. Important is from the beginning to specify what represents the perinatal period (according to your understanding), this is very relevant information, currently missing, that might guide you in structuring of the manuscript.
Thanks for the suggestion, we consider that is an important issue and appreciate the comment. To perform this review we considered the perinatal period which starts at the 20th to 28th week of gestation and ends at 4 weeks after birth as well as infant period to around 1 year after birth. Is spite of some authors consider infants until 2 year from birth, we focused in infants with 1 year of age.
In Material and Methods sections, we included some sentences in order to clarify these points (Lines 86 to 91):
“A bibliographic search strategy was conducted to identify all studies reporting on the use of probiotics during pregnancy, and the neonatal period and the infant period, highlighting their impact on neonatal and fetal and neonatal infant health. Perinatal period starts at the 20th week of gestation and ends 4 weeks after birth including pregnancy and neonatal period. Furthermore infant period extends from birth to one year of age. However, studies in infants with a treatment extended until two years of age were also considered in the analyses of the present review in order to evaluate the impact of the probiotic treatments”.
2. It is clear that medication impact the microbiome, especially those with oral intake. Two comments in this aspect, first as presented here it is not clear how this relates to dysbiosis in the perinatal period, and second did the author considered into their selection criteria for this point to exclude studies where the drugs were not delivered orally? Also, the respective table (Table 1) is very confusing as it also includes both intervention with preterm infants or studies on the impact of drugs in postnatal period (more than a week).
Thanks for the suggestions. Regarding the first comment, the perinatal period (as we have indicated in the manuscript according to the reviewer's comments) starts at the 20th to 28th week of gestation and ends, approximately, at 4 weeks after birth. The data collected in the table belong to populations amongst 32-36 weeks of gestation (preterm) and full term neonates which are subjected to antibiotics during the first weeks of life (never beyond the first month of life). Therefore, exposure to antibiotics occurs during the perinatal period.
Regarding the second question, we analyzed the antibiotic use during pregnancy or intrapartum in order to evaluate the impact on early microbial colonization. Taking into account that the antibiotics administered to the mother can reach the newborn via the umbilical cord (Pacifici GM, 2006) or alter early colonization by altering the mother's intestinal and vaginal microbiota (Muellet NT et al., 2015), we decided to include these groups to the analysis (prenatal exposure and intrapartum exposure).
Pacifici GM. Placental transfer of antibiotics administered to the mother: a review. International journal of clinical pharmacology and therapeutics. 2006;44(2):57-63.
Mueller NT, Bakacs E, Combellick J, Grigoryan Z, Dominguez-Bello MG. The infant microbiome development: mom matters. Trends Mol Med. 2015;21(2):109-17.
An early antibiotic treatment is common among premature infants given their significant risk of early-onset sepsis. Moreover, between 20% and 30% of women receive intrapartum antibiotic prophylaxis to prevent sepsis in infants (Tapiainen T., 2019). For that, we added both preterm infants undergoing antibiotics and neonates to the study. In summary, all the moments during the perinatal stage in which the individual may be undergoing antibiotic therapy (prenatally, during delivery or during the few days after birth) were selected.
Tapiainen, T., Koivusaari, P., Brinkac, L. et al. Impact of intrapartum and postnatal antibiotics on the gut microbiome and emergence of antimicrobial resistance in infants. Sci Rep 9, 10635 (2019).
We have also modified some points of the table to make it more understandable for the reader. We have added a new column in which we establish the time of exposure to antibiotics (pre-postnatal, postnatal or during delivery). Furthermore, we have divided the table 1 into two sections, preterm infants and full term infants in order to clarify this issue.
Regarding the administration of the antibiotic, unless otherwise indicated, it is usually given intravenously. In our analysis, we did not exclude studies where the antibiotic was administered by this route since it is largely used. Additionally, the route of administration is rarely indicated in the papers since it is commonly chosen.
For example, the Canadian, UK and American guidelines use some antibiotic orally (erythromycin) in premature rupture of membranes in the preterm period. Although each geographical area adapts these guidelines depending on the local microbiology. The guides I refer are the following:
1) ACOG Practice Bulletin No. 199: Use of Prophylactic Antibiotics in Labor and Delivery, Obstetrics & Gynecology: September 2018 - Volume 132 - Issue 3 - p e103-e119
2) Yudin MH, van Schalkwyk J, Van Eyk N. No. 233-Antibiotic Therapy in Preterm Premature Rupture of the Membranes. J Obstet Gynaecol Can. 2017;39(9):e207-e212.
3) Hughes RG, Brocklehurst P, Steer PJ, Heath P, Stenson BM on behalf of the Royal College of Obstetricians and Gynaecologists. Prevention of early-onset neonatal group B streptococcal disease. Green-top Guideline
No. 36. BJOG 2017;124:e280–e305.
3. Similarly, to the previous most of the information, with small exceptions (Ref 88), presented in the section 3.2. Early Aberrant Microbiota and its effect on pediatric disease, is difficult to link to the general aim.
As the reviewer has suggested, this point has been rewritten according to the manuscript storyline. Paragraphs whose information was not particularly relevant have been removed and new ones have been added to help a better understanding of the section.
We consider this section useful to give context to the following sections, updating the recent research in the field. After discuss about the prenatal development of the microbiota and the effects of antibiotics used during childbirth or in NICUs, we introduce the role of the microbiota in the maturation of the immune system and how dysbiosis can imbalance it (section 3.3). In addition, we show the importance of Bifidobacteria in Th1 / Th2 balance and their role on gut barrier. At the end of the section we present some examples of the correlation between certain diseases (such as atopic dermatitis or asthma) with certain microbial groups. Once we have presented the problem, the manuscript introduces the probiotics and their use during the prenatal and postnatal period as a preventive element in different allergic diseases.
Therefore, although this point may present some general information, we consider that it is necessary to emphasize the importance of the microbiota in the maturation of the immune system and the effects that a dysbiosis can generate in health. This section also gives us the opportunity to present the use of probiotics as a preventive tool.
4. Use research papers and less review papers, e.g. line 237 – Ref 53 and 53, line 278-283 is justified with only one reference (Ref 73), which all are actually review papers. Moreover, general statements as for examples those at line 143-144, line 166-167 need to be supported with respective references.
As the reviewer recommends we have replaced the review papers to original researches:
Line 237 (Ref 52 and 53): Now this paragraph is located in line 244, review references have been deleted and replaced. Thus the paragraph is as follows: “……..promoting harmful consequences such as necrotizing enterocolitis (NEC) (Weintraub AS et al. 2012; Alexander et al. 2011; Kuppala et al. 2011) or fungal infections as candidemia (Benjamin DK et al. 2003).
Line 278-283 (Ref 73): Now this paragraph is located in line 297, the review reference has been deleted, new research references have been added and some phrases have been added for a better understanding. Thus the paragraph is as follows: “……Allergic diseases occur at any stage of life, although some allergic manifestations, such as allergies to food, are most likely to develop during the first few years of life (Gupta E et al. 2011). After three years of age, prevalence of IgE specific to inhalant allergen become predominant (Bruno G, et al. 1995). Several variables as antibiotic consumption during pregnancy, mode of delivery, feeding and mother’s lifestyle during pregnancy are key factors that will shape de neonatal gut microbiome (Stearns JC et al 2017; Stokholm J et al. 2014; Jakobsson HE et al. 2014 and Chu Dm et al. 2016).
Line 143-144: Now this paragraph is located in line 142, a research reference has been added. Thus the paragraph is as follows: “ Human microbiome colonization can be understood as a progressive process. In puberty and adulthood, the microbiota shows a higher diversity than in newborns (Nagpal, R.,et al 2018)”.
Line 143-144: Now this paragraph is located in line 166, a research reference has been added. Thus the paragraph is as follows: “It is widely accepted that the early neonatal gut microbiome comes from maternal strains (Korpela, K., et al. 2018)”.
Please note that we have also added new references regarding the prevalence of food allergies at the beginning of section 3.5, removing the reference from a review. Thus the paragraph is as follows: “Food allergies (FA) have become a common problem that affects approximately 6% of infants under two and 9% of children aged 3 to 5 (Gupta, R, et al.2011). Eggs, milk and peanuts are the most common food allergens and skin problems as eczema are closely associated with FA (Eigenmann PA 1998; Eigenmann PA et al. 2000).
5. Not agree that neither Prevotella (line 171) nor Clostridium coccoides (line 182-183) are pathogens.
The mention about the possible pathogenesis of Prevotella comes from the study cited in the manuscript (Li et al. 2019): “Other differential bacteria such as Prevotella, Staphylococcus, Escherichia/Shigella and other opportunistic pathogens showed high relative abundance in the disinfection group”. Although is not common, some Prevotella strains may be pathobionts that can participate in human disease by promoting chronic inflammation (Larsen JM. The immune response to Prevotella bacteria in chronic inflammatory disease. Immunology. 2017;151(4):363-374.). Although as the reviewer comments, Prevotella is generally considered a commensal, common in non-Westernised populations consuming plant-rich diet. Therefore, to avoid misunderstanding we have removed that genus of the paragraph, remaining as follows:. “Their results showed a lower presence of Lactobacillus and more opportunistic pathogens such as Staphylococcus, Klebsiella, and Escherichia in the disinfected group compared to the nondisinfected and C-section groups”.
Regarding Clostridium coccoides, we have reviewed the bibliography and we totally agree with the reviewer and we appreciate the comment. We have rewritten the paragraph being as follows: “In relation to breastfeeding, some authors revealed that breastfed children had a high presence of Bifidobacterium in their gut and a lower abundance of Clostridiales (Hesla 2014). Conversely, formula-fed infants showed fewer bifidobacteria and had significantly higher proportions of Bacteroides and members of the Clostridium coccoides and Lactobacillus groups (Fallani 2010).”
6. Another issue is related to the statement related to the exclusion criteria for the included studies. Mentioned in line 109 that interventions using probiotics with prebiotics were excluded. However, if you follow the studies summarized in Table 2 – Ref 117 where the probiotic mixture was provided to the infants mixed with syrup and GOS, the last is a prebiotic. In the studies summarized in Table 3, Ref 125 – probiotic was mixed with inulin (a well-known prebiotic); Ref 138 – probiotic was provided to the infants in mix with GOS (prebiotic).
The reviewer is right, it was a mistake. We clarify this point indicating that (Lines 113-114) “. Interventions using only prebiotics or immunotherapy were also excluded (see details in the Supplementary Materials: Methodology).” Studies based on a combined treatment of prebiotics and probiotics are included in tables 2, 3 and 4.
7. The reviewer suggests the authors to pay a better attention on what microbiota has been investigated in certain reference and how it fits with their report. In the reference discussed in line 169-172 – Ref 28, the neonatal oral microbiome has been investigated. How this fit with the Gut colonization section?
The authors appreciate this comment and we agree. We have expanded this paragraph with a new bibliography on the latest advances in the origin of the oral microbiota in neonates that will reach the intestine to help its development.
Therefore, the new paragraph is as follows: “The first colonizing bacteria enter the intestine through the oral cavity. Although it was thought that the birth canal microbiota was the greatest modulator of infants’ oral and gut microbiota, recent studies suggest that neonatal oral cavity might have a prenatal origin preceding exposure to the birth canal (Tuominen, H., Collado, M. C., Rautava, J., Syrjänen, S., & Rautava, S. (2019)). Although it has been shown that an imbalance in the birth canal can affect the neonate's oral microbiota. Li et al. studied whether vulvar disinfection with povidone iodine had an effect on neonatal’s oral microbiota in 30 infants. Their results showed a lower presence of Lactobacillus and more opportunistic pathogens such as Prevotella, Staphylococcus, Klebsiella, and Escherichia in the disinfected group compared to the nondisinfected and C-section groups”.
8. Regarding the sections 3.3 and 3.4 that are actually an important part of the manuscript, need to be revised to fit with the perinatal period. Number of studies reported in the respective tables include report data for the postnatal period. The current tables are very busy, removing the postnatal data will make the tables more clear.
We appreciate the reviewer's comments. The purpose of these sections (now 3.4 and 3.5) is to compare prevention in certain allergies according to whether the probiotics are given only during pregnancy, only after birth or in both stages. Thus, three groups appear, prenatal, postnatal and both. Most studies in both pre- and post-natal, the probiotic is given from birth to 6 months of age. No studies with administration beyond one and a half years of age have been included. Except for the one conducted by Wickens (2018) because of its importance, since it is the study with the longest follow-up to date. Thus, these sections include both a perinatal and infant period (up to one year of age). As the reviewer has rightly commented, we have included in the text the difference between the two stages to avoid this type of misunderstanding in the manuscript.
Considering that one of the objectives of these two sections is to see if there is a differential effect in allergy prevention depending on whether the probiotic is administered before birth, after birth, or both stages, we think it would be a shame to eliminate the postnatal group as it is an important group in the comparison.
We appreciate the indication on the tables, we have reviewed and rewritten them, eliminating redundant information and focusing on key information. In addition, we have divided each table into two, so that studies in the postnatal stage are indicated in a separate table so that the reader can follow the information in a more orderly and easy way.
9. Keep all the table in the same style – the style of Table 1 and Table 2 is different than that of Table 3 and Table 4.
Reviewer is right. All the tables have been modified to share the same style.
10. It is necessary to describe all abbreviations used in the text and the tables as well.
As reviewer suggested, the following abbreviations have been included in the text or the tables:
C-section: Caesarean section
- longum: Bifidobacterium longum
- catenulatum: Bifidobacterium catenulatum
spp.: several species
subsp: subspecie
LGG: Lactobacillus rhamnosus Gorbach-Goldin (tables 2,3,4)
CFU: colony-forming unit (tables 1,2,3,4)
RCT: randomized clinical trials
- epidermidis: Staphylococcus epidermidis
11. There is need to improve the numbering of the sections. At the moment there are two sections with number 3.1 – Prenatal development of the microbiome and early colonization; and Gut dysbiosis induced by antibiotic and nonantibiotic medications. Also, not clear why there is a 3.1.1. Early colonization.
Thanks for the comment, our mistake. We corrected the numbering of 3.2 Gut dysbiosis induced by antibiotic and nonantibiotic medications. Moreover, the head of the section 3.1.1 Early colonization has been deleted.
Reviewer 2 Report
The recent interest in the efficacy of probiotics has attracted a lot of attention. Similar reviews have been published a lot for the past several years. This review did not contain any meta-analysis due to the discrepancy of the study protocols which also leads to the fact that authors’ opinion will determine the conclusion in this manuscript. Recently, early fetal stage lung colonization has been reported by University of Alabama at Birmingham which stir up even more debate about whether the amniotic fluid is sterile or not since the meta/megagenomic study show bacterial RNA can be detected as early as 6 weeks of gestation. I will not be surprised that studies in full term infants usually will not get significant difference since most of them are not sick and will not be exposed to prolonged antibiotic treatment. What clinicians are more interested will be the high-risk premature infants who have higher percentage of morbidities and will be easier to find a statistically significant difference. I have to commend that the authors have attempted to do their best to summarize the findings but the information still very conflicting to the best. The inclusion of skin microbiome in the review probably will not provide new information to the readers since it will be more environmentally determined for those NICU infants. The skin microbiome similar to the spectrum in breast milk is not a surprise since no mother will sterilize the nipples before pumping. It is hard to fathom the premature infant admitted to NICU can be breastfed. Were they breastmilk-fed instead? I will not be surprised that antibiotics will impact the neonatal gut microbiota. The authors described three different types of antibiotics treatments including IAP, IAP+postnatal antibiotics, and postnatal antibiotics alone. The authors have the responsibility to summarize the association of each type of antibiotics treatment and the change in gut microbiota instead of letting the readers to figure them out. The association between gastroenteritis and future allergic disease remains unsettled. On the contrary, children in developing countries exposing to relatively less hygienic environment are more commonly suffer from gastroenteritis but with much less atopic diseases as compare to children in developed countries where prevalence of atopic disease has been increasing although. The enteromammary pathway is another debatable mechanism. Recent studies also suggest a possible retrograde colonization. I have to agree that the possibility remains unsettled. There are a few minor concerns required to be addressed.
In lines 44-45: For that, the standard profile of healthy infant microbiota is the fecal microbiota of full term infants, vaginally delivered and exclusively breastfed. It is true but the authors need to be aware that maternal diet and habit will significantly influence the maternal gut microbiota that can lead to regional difference.
In lines 65-66: “asthma are directly related to the balance between T-helper type 1- (Th1) and Th2-associated chemokines.” I think cytokines will be better than chemokines since the later only applies to proteins that attract cell to move while the former has more broad physiological coverage.
In lines 69-70: The authors described “Prenatally, probiotics are mainly transferred from mother to
infant by breastfeeding.” I have a practical question about how the mothers breastfeed the fetuses in utero? I have practiced for more than 30 years and would like to know how this creative feeding achieved? This type of feeding will be a big scientific breakthrough!
In lines 69-70: “Dysbiosis triggered by these imbalances in the microbiota generates a severe delay of immune system maturation, which produces autoimmune and atopic diseases such as asthma, rhinitis [8], and lifelong food sensitization [9].” I do not think we have definite, direct, proof that dysbiosis causes autoimmune and atopic diseases at this moment. Studies have shown a strong association or suggest an association.
In lines 216-217: “However, infants with posteriorly 216 developed chronic lung disease showed reduced bacterial diversity at birth [47].” I do not understand why “posteriorly”?
In lines 266-268: Therefore, a restricting antibiotic stewardship may be of utmost importance to reduce unnecessary and harmful 267 antibiotic consequences.” The word “restricting” only adds confusion to the sentence and can be removed. Antibiotic stewardship already means a policy of judicious use of antibiotics.
In lines 283-286: “The capacity to initiate the Th1 response is 283 particularly limited in neonates and infants as they are partially protected by placental transfer of 284 maternal IgG during the last stages of pregnancy (Kachikis and Englund, 2016), and by secretory IgA 285 (SIgA) in breast milk if breastfed.” (Kachikis and Englund, 2016) can be put into the reference list.
In line 605: “---all of them clinical trials---”. Minor grammatical error needs to be corrected.
Author Response
The recent interest in the efficacy of probiotics has attracted a lot of attention. Similar reviews have been published a lot for the past several years. This review did not contain any meta-analysis due to the discrepancy of the study protocols which also leads to the fact that authors’ opinion will determine the conclusion in this manuscript. Recently, early fetal stage lung colonization has been reported by University of Alabama at Birmingham which stir up even more debate about whether the amniotic fluid is sterile or not since the meta/megagenomic study show bacterial RNA can be detected as early as 6 weeks of gestation. I will not be surprised that studies in full term infants usually will not get significant difference since most of them are not sick and will not be exposed to prolonged antibiotic treatment.
Thank you for the comment. As reviewer indicated, the methodology of meta-analysis is hard to adapt in our review process, due to not only by the heterogeneity of the issues addressed but also by the different protocols performed in the papers. We followed the PRISMA methodology to try to expose the results and conclusion objectively. However, we also thought that a mention of our more valuable, relevant and analyzed trends for the distinct issues, once we reviewed the papers in deep, would be valuable for the readers.
We consider a great input the paper mentioned for the reviewer so that we have included the reference in the introduction section as follows “Recent studies have indicated that microbiota colonization of the human body starts during pregnancy, altering the paradigm of the fetus as a sterile organism [1-3]” (reference 1 in Bibliography). In reference of studies in healthy full term infants, we agree to the reviewer since most of the studies presented in this review did not get significant results among groups, except in preterm infants or after an antibiotic treatment.
What clinicians are more interested will be the high-risk premature infants who have higher percentage of morbidities and will be easier to find a statistically significant difference. I have to commend that the authors have attempted to do their best to summarize the findings but the information still very conflicting to the best. The inclusion of skin microbiome in the review probably will not provide new information to the readers since it will be more environmentally determined for those NICU infants. The skin microbiome similar to the spectrum in breast milk is not a surprise since no mother will sterilize the nipples before pumping. It is hard to fathom the premature infant admitted to NICU can be breastfed. Were they breastmilk-fed instead? I will not be surprised that antibiotics will impact the neonatal gut microbiota.
The authors appreciate the comments of the reviewer, certainly the skin microbiota is variable and depends on the newborn´s environment. We consider that we cannot ignore that the characteristics of skin microbiota in newborn could result in healthcare associated infections, for this reason we have mentioned it, but we have not go deeper into the topic because it was not the scope of the article. Regarding the paper of Soeorg et al. 2017, the authors showed that the skin and gut of preterm neonates is colonized with S. epidermidis that is distinct from strains found in breast milk, but gradually the gut is enriched with strains genetically similar to those in breast milk, as in term neonates.
Indeed, the preterms were fed with breast milk, as indicated in the material and methods of the paper: “Between January 2014 and December 2015, we recruited preterm neonates (gestational age (GA) o37 weeks) and their mothers if feeding with the mother’s own unpasteurized BM was initiated within the first week of life, from the third-level neonatal intensive care units (NICUs) of Tallinn Children’s Hospital or Tartu University Hospital”. Those preterms that tolerate it can be fed breast milk. The mother expresses the milk with a breast pump and then it gives to the preterm through a fine feeding tube or with a syringe.
The authors described three different types of antibiotics treatments including IAP, IAP+postnatal antibiotics, and postnatal antibiotics alone. The authors have the responsibility to summarize the association of each type of antibiotics treatment and the change in gut microbiota instead of letting the readers to figure them out.
As reviewer suggests, we included a new paragraph indicating the changes of phyla produced by the different antibiotic treatments.
Lines 268-276 “Moreover, dybiosis induced by antibiotics was also analyzed detecting more studies which assessed IPA rather than a postnatal antibiotic therapy. In the IPA group the decrease in Bifidobacteriaceae colonization was the most common dysbiotic microbiome alteration [64-67,73-75,79] followed by Proteobacteria increase [63,65-68,73], Bacteroidetes decrease [63,67,68,73,75], and Actinobacteria decrease [63,65-67,75]. Increased Enterobacteriaceae colonization was reported in the IPA group of three studies [63,65,75]. Postnatal antibiotic therapy was associated with increased Enterobacteriaceae colonization [10,70,72-74] followed by Proteobacteria increase [69,70,72,73] and Bifidobacteriaceae decrease [70,71,73,74]. Bacteroidetes [69,70,72] and Actinobacteria decrease [70] were less frequently reported.”
The association between gastroenteritis and future allergic disease remains unsettled. On the contrary, children in developing countries exposing to relatively less hygienic environment are more commonly suffer from gastroenteritis but with much less atopic diseases as compare to children in developed countries where prevalence of atopic disease has been increasing although. The enteromammary pathway is another debatable mechanism. Recent studies also suggest a possible retrograde colonization. I have to agree that the possibility remains unsettled. There are a few minor concerns required to be addressed.
We agree with the reviewer's opinion. The study by Hui-Hsien Pan et al. (2019) although it was a retrospective cohort including large sample size (94,929 children) and follow-up for 5 years has some limitations. No clinical evaluation or investigation of biological samples or the non-evaluation of family history and environmental effects makes the study, although enormously interesting and novel, require more solid data. Therefore, we have added the following sentence to the manuscript (line 334-336) to summarize the indicated, remaining as follows: “However, the authors did not performed a clinical evaluation of biological samples, thus further studies to find the association between early-infectious-GE, early-noninfectious-GE, and allergic disease are needed”.
Regarding the second point, the “old friends mechanism” suggests that early and regular exposure to harmless microorganisms—“old friends” present throughout human evolution- interact with the regulatory systems that keep the immune system in balance and prevent overreaction, which is an underlying cause of allergies. As reviewer says it is totally true that children in developed countries have higher allergies than children in developing countries due to the decrease of certain key bacterial groups for the correct maturation of the immune system. For example, there are less lactobacilli in the guts of children with allergies (Björkstén B, Naaber P, Sepp E, Mikelsaar M Clin Exp Allergy. 1999 Mar; 29(3):342-6). We have expanded and detailed the information about the effect of the microbiota on the maturation of the immune system and the importance of a correct microbiological balance to avoid exacerbated Th2 responses in the future. Please find all this information in the section 3.3 “early aberrant microbiota and its effect on pediatric diseases”.
Regarding the entero-mammary pathway, we are agree with the reviewer that there are two hypotheses regarding the origins of milk microbiota: the “retrograde transfer” of external bacteria, and the “entero-mammary pathway” for translocation of internal bacteria. Recent findings related with milk microbiota composition suggests the exogenously derived bacteria have a stronger role in milk inoculation than the entero-mammary pathway. Although the scope of section 3.4 is to explain the transfer of probiotics from mother to offspring, not the transfer of the microbiota itself, so we decided to focus on the entero-mammary route. The translocation of probiotics from the gut occurs more frequently during the late stages of pregnancy due to altered tight junction regulation in the intestinal tract. Evidence for the entero-mammary pathway in probiotics is the presence of orally administered probiotic strains in the milk of lactating mothers in both human and animal studies:
- Treven P, Mrak V, Bogovic Matijasic B, Horvat S, Rogelj I. Administration of probiotics lactobacillus rhamnosus GG and lactobacillus gasseri K7 during pregnancy and lactation changes mouse mesenteric lymph nodes and mammary gland microbiota. J Dairy Sci. 2015;98:2114–2128. doi:10.3168/jds.2014-8519.
- Abrahamsson TR, Sinkiewicz G, Jakobsson T, Fredrikson M, Björkstén B. Probiotic lactobacilli in breastmilk and infant stool in relation to oral intake during the first year of life. J Pediatr Gastroenterol Nutr. 2009;49:349–354. doi:10.1097/MPG.0b013e31818f091b.
- de Andres J, Jimenez E, Chico-Calero I, Fresno M,Fernandez L, Rodriguez JM. Physiological translocation of lactic acid bacteria during pregnancy contributes to the composition of the milk microbiota in Nutrients. 2018;10:14. doi:10.3390/nu10010014
Therefore, we appreciate the comments of the reviewer and following their suggestions we have mentioned the retrograde transfer and its importance in the composition of breast milk in line 355: “Moreover, retrograde transfer of external bacteria into the mammary gland has also a strong role in milk inoculation during lactation [117].”.
In lines 44-45: For that, the standard profile of healthy infant microbiota is the fecal microbiota of full term infants, vaginally delivered and exclusively breastfed. It is true but the authors need to be aware that maternal diet and habit will significantly influence the maternal gut microbiota that can lead to regional difference.
Following author recommendation, we included that sentence to clarify the point in Lines 46-48 “However, it is important to mention that maternal diet and habits will significantly influence maternal gut microbiota which can lead to regional differences.”
In lines 65-66: “asthma are directly related to the balance between T-helper type 1- (Th1) and Th2-associated chemokines.” I think cytokines will be better than chemokines since the later only applies to proteins that attract cell to move while the former has more broad physiological coverage.
We changed the term to cytokines.
In lines 69-70: The authors described “Prenatally, probiotics are mainly transferred from mother to infant by breastfeeding.” I have a practical question about how the mothers breastfeed the fetuses in utero? I have practiced for more than 30 years and would like to know how this creative feeding achieved? This type of feeding will be a big scientific breakthrough!
You are absolutely right! Sorry for that, our mistake. We changed to postnatally.
In lines 69-70: “Dysbiosis triggered by these imbalances in the microbiota generates a severe delay of immune system maturation, which produces autoimmune and atopic diseases such as asthma, rhinitis [8], and lifelong food sensitization [9].” I do not think we have definite, direct, proof that dysbiosis causes autoimmune and atopic diseases at this moment. Studies have shown a strong association or suggest an association.
Absolutely agree. We modified the sentence as follow “Dysbiosis triggered by these imbalances in the microbiota generates a severe delay of immune system maturation, which can lead to autoimmune and atopic diseases such as asthma, rhinitis [8], and lifelong food sensitization [9].”
In lines 216-217: “However, infants with posteriorly 216 developed chronic lung disease showed reduced bacterial diversity at birth [47].” I do not understand why “posteriorly”?
As reviewer recommended we changed the sentence in order to clarify its meaning, deleting the word “posteriorly” as follow: infants which developed chronic lung disease showed…
In lines 266-268: Therefore, a restricting antibiotic stewardship may be of utmost importance to reduce unnecessary and harmful 267 antibiotic consequences.” The word “restricting” only adds confusion to the sentence and can be removed. Antibiotic stewardship already means a policy of judicious use of antibiotics.
Following reviewer recommendation we have deleted the word restricting.
In lines 283-286: “The capacity to initiate the Th1 response is 283 particularly limited in neonates and infants as they are partially protected by placental transfer of 284 maternal IgG during the last stages of pregnancy (Kachikis and Englund, 2016), and by secretory IgA 285 (SIgA) in breast milk if breastfed.” (Kachikis and Englund, 2016) can be put into the reference list.
Thanks for the comment. We added the reference Kachikis and Englund, (2016) into the reference list.
In line 605: “---all of them clinical trials---”. Minor grammatical error needs to be corrected.
According to reviewer suggestion, we changed the sentence to “all clinical trials”.
Round 2
Reviewer 1 Report
The reviewer appreciates the efficient revision of the manuscript. All comments are very well addressed, the revised manuscript reads better. Well done!